



# Statistical evaluation of methane isotopic signatures determined during near-source measurements

Sara M. Defratyka[1,2,3], James L. France[4,5,6], Rebecca E. Fisher[5], Dave Lowry[5], Julianne M. Fernandez[5,7], Semra Bakkaloglu[5,8], Camille Yver-Kwok[1], Jean-Daniel Paris[1], Philippe Bousquet[1], Tim Arnold[2,3], Chris Rennick[2], Jon Helmore[2], Nigel Yarrow[2], Euan G Nisbet[5]

[1]Laboratoire des Sciences du Climat et de l'Environnement (LSCE-IPSL) CEA-CNRS-UVSQ Université Paris Saclay, France
[2]National Physical Laboratory (NPL), UK
[3]School of GeoSciences, University of Edinburgh, Edinburgh, UK
[4]British Antarctic Survey, Natural Environment Research Council, UK
[5]Department of Earth Sciences, Royal Holloway, University of London, UK
[6]Environmental Defense Fund, UK
[7]Department of Geology, University of Maryland, College Park, Maryland, USA
[8]Sustainable Gas Institute, Imperial College London, UK

*Correspondence to*: Sara M. Defratyka (sara.defratyka@ed.ac.uk)

**Abstract.** Stable carbon isotopic signatures of methane emissions are broadly used for methane source identification, apportionment, and global-scale modelling of methane sources and sinks. Thus, accurate and precise isotopic measurements of methane are crucial for methane studies from the local to global scale. To answer the need for robust and verified measurement methods, we aim at defining the best practice to determine isotopic signatures of methane sources, considering accessibility, practicality, costs, accuracy, and precision. Using Keeling and Miller-Tans methods, we verify the impact of linear fitting methods, averaging approaches, and, for Miller-Tans method, differently defined backgrounds. Verification is carried out for measurement sets using Isotope Ratio Mass Spectrometry and Cavity Ring Down Spectroscopy (CRDS). The use of AirCore for sampling, with subsequent measurements by CRDS, is also examined. Different analytical strategies introduce bias in determining isotopic signatures of methane sources, and the crucial role of rejection criteria is demonstrated. Overall, the most robust results are obtained for non-averaged data using fitting methods, which include uncertainties on x- and y-axis values.

## 1. Introduction

To better understand the global $CH_4$ budget, additional tracers, such as alkanes (e.g. ethane) or stable isotopic signatures, can be measured alongside the $CH_4$ mole fraction, as they provide additional information about $CH_4$ source apportionment (e.g. Simpson et al. 2012; Rella et al. 2015; Sherwood et al. 2017; Turner, Frankenberg, and Kort 2019; Basu et al. 2022). Typically, stable carbon isotopic signatures of methane emissions (expressed as $\delta^{13}CH_4$) measurements are widely used, from local to global scales to characterise emission sources from individual sites to better constrain $CH_4$ budget changes



(Phillips et al., 2013; Rella et al., 2015; Röckmann et al., 2016; Lopez et al., 2017; Hoheisel et al., 2019; Maazallahi et al., 2020; Menoud et al., 2020, 2021; Defratyka et al., 2021; Al-Shalan et al., 2022). However, $\delta^{13}CH_4$ values of individual
methane sources from one sector (e.g. landfill, natural gas) vary globally, depending on numerous factors, like formation processes, locations, or management (e.g. Whiticar 1999; Chanton et al. 2000; Sherwood et al. 2017; Menoud et al. 2022). Moreover, $\delta^{13}CH_4$ of some sectors are spread across a wide range and overlap with $\delta^{13}CH_4$ for other sectors (e.g. Menoud et al. 2022; Sherwood et al. 2017; Fernandez et al. 2022). Therefore, a better understanding of $\delta^{13}CH_4$ source signatures can improve source attribution in top-down emission studies (atmospheric observation combined with the inverse modelling),
(e.g. Saunois et al. 2020; Varga et al. 2021; Basu et al. 2022) to verify emissions from bottom-up approaches, which use process-based models, inventories and data extrapolation (Rigby et al., 2012; Schwietzke et al., 2016; Lan et al., 2021).

Regarding individual sectors, (e.g., natural gas, agriculture, landfill), $\delta^{13}CH_4$ can be measured in the atmosphere in near-source conditions (i.e., downwind of a $CH_4$ source). In this case, isotopic signatures can be sampled from ambient air by taking bag/canister samples and measured afterward in the laboratory (e.g. Townsend-Small et al. 2012; 2016; Lowry et al.
2020; Bakkaloglu et al. 2021; 2022;). An alternative is to deploy in-situ instruments, for example, a Cavity Ring Down Spectroscopy (CRDS) instrument equipped with an AirCore (air storage tool) (Karion et al., 2010; Rella et al., 2015) to increase sampling precision (Lopez et al. 2017; Hoheisel et al. 2019; Defratyka et al. 2021). Calculating a source's $\delta^{13}CH_4$ signature is complicated by 'background' air, i.e. the atmospheric air that exists before mixing and being influenced by a source. To extract background δ13CH4 from the near-source ambient air samples, a Keeling method (Keeling, 1961; Pataki
et al., 2003) or Miller-Tans method (Miller and Tans 2003) can be used. These methods are based on the principle of mass balance conservation. Both methods use a linear regression to determine $\delta^{13}CH_4$ methane sources. As such, the calculation method of choice has an impact on determining a source's isotopic signature and can potentially bias determined $\delta^{13}CH_4$ (Miller and Tans 2003; Zobitz et al. 2006; Wehr and Saleska 2017).

To the best of our knowledge, the verification of the use of Keeling and Miller-Tans methods to determine $\delta^{13}CH_4$ from
near-source measurements has never been conducted under controlled and realistic field conditions. To fill this gap and better understand these methods, as well as derive a more universal approach, isotopic measurement and sample collection were included within a controlled release experiment. The experiment focused on the methods applied during mobile, vehicle-based methane measurements. Samples collected over five days of the experiment were used to compare Isotope Ratio Mass Spectrometry (IRMS) and CRDS measurement techniques. Moreover, the studies were focused on a
comprehensive inter-comparison of Keeling and Miller-Tans methods, including the impact of averaging clusters, and for Miller-Tans method, the impact of chosen backgrounds. Finally, data were re-analysed using different linear fitting methods.

As $\delta^{13}CH_4$ measurements are now widely used in understanding atmospheric methane, both on source (Menoud et al., 2022) and global scale (Nisbet et al., 2019), improved determinations of $\delta^{13}CH_4$ source signatures could refine the constraint to infer $CH_4$ source distributions from regional to global scales. The measurement and data analysis techniques developed in
this study could also be useful for those studying additional problems in greenhouse gas and carbon cycle science by





improving the understanding of the contribution of different emission sources. We also expect the result to generalize to other applications beyond mobile measurements of methane, such as continuous time series studies.

## 2. Controlled release experiment and sampling methodology

### 2.1. Controlled release set up

The controlled release experiment allows an evaluation of the accuracy and precision of mobile near-source measurements of $CH_4$ emission rate, $C_2H_6:CH_4$ and $\delta^{13}CH_4$ under strictly supervised conditions. The experiment lasted over 5 days in September 2019 at Bedford Aerodrome, UK. Pure methane was released from a manifolded multi-cylinder pack, of twelve cylinders containing $999.6 \pm 10.0$ mmol mol$^{-1}$. The impurities in cylinders came from ethane ($48 \pm 10$ µmol mol$^{-1}$) and propane ($0.149 \pm 0.30$ µmol mol$^{-1}$). The methane release rate varied up to 70 L min$^{-1}$. During the release, $CH_4$ was mixed

with ethane ($C_2H_6$) in a varying ratio, giving $C_2H_6:CH_4$ ratios from 0.00 to 0.07. The purity of the $C_2H_6$ was $999.9 \pm 10.0$ mmol mol$^{-1}$, with impurities mostly from methane ($2.27 \pm 0.46$ µmol mol$^{-1}$) and propane ($7.5 \pm 1.5$ µmol mol$^{-1}$). The details of the experimental setup configuration, particularly about gas blending and control centre can be found in Gardiner et al. 2017. All 12 cylinders were filled at the same time from the same $CH_4$ source, ensuring $\delta^{13}CH_4$ remained stable over the entire measurement period. Overall, the controlled release experiment involved 24 releases, each lasting about 45 minutes.

Throughout the paper, the units of ‰ represent $\delta^{13}CH_4$ and are not an indication of relative error in the results.

### 2.2. Direct sampling from cylinder batch to determine $\delta^{13}CH_4$

To directly determine $\delta^{13}CH_4$ of the source gas, a sample cylinder was filled directly from the multi-cylinder pack after the end of the experiment. Then, the sample was diluted to approximately 600 µmol mol$^{-1}$ and measured using laser spectrometry (Rennick et al., 2021). In the next step, 600 µmol mol$^{-1}$ sample was diluted to 2.5 µmol mol$^{-1}$ and measured

using IRMS at Royal Holloway, University of London (Fisher et al., 2006). $\delta^{13}CH_4$ measured by laser spectrometry is equal to $-41.45 \pm 0.06$ ‰ (1 Standard Deviation - 1SD), while $\delta^{13}CH_4$ measured by IRMS achieved $-41.27 \pm 0.06$ ‰ (1SD) (Rennick et al., 2021). Direct measurements for $\delta^{13}CH_4$ by laser spectrometry and IRMS are compatible within 2SD and was used as a true $\delta^{13}CH_4$ signature of the multi-cylinder pack. The true $\delta^{13}CH_4$ signature was compared with results from samples collected using mobile systems, described in section 2.3.

### 90 2.3. Mobile sampling methodology set-up

The controlled release experiment gave the opportunity to validate the mobile laboratories of Royal Holloway, University of London (RHUL) and the Laboratory for Sciences of Climate and Environment (LSCE). The RHUL mobile laboratory used for this experiment was in operation between 2013 and 2020 (Lowry et al., 2020). This vehicle was equipped with a Picarro CRDS G2301 analyser for $CH_4$ mole fraction measurements, a Los Gatos Research Ultraportable Methane Ethane Analyzer



(LGR UMEA) and a manually operated diaphragm pump for air sample bag filling. Three air cylinders were measured and calibrated against the NOAA scale by the Max-Plank Institute for Biogeochemistry Jena, which were used to calibrate the Picarro G2301 before and after the measurement campaign to the WMO X2004A $CH_4$ scale (Lowry et al., 2020; France et al., 2016; Zazzeri et al., 2015).

The LSCE mobile laboratory was previously used during mobile studies (Defratyka et al. 2021), and it is similar to other
mobile laboratories equipped with a Picarro CRDS G2201-I (henceforth referred to as CRDS), capable of in-situ measurements of $CH_4$ mole fraction and $\delta^{13}CH_4$ (e.g. Rella et al. 2015; Lopez et al. 2017; Hoheisel et al. 2019). The mobile set-up of LSCE is supplied with an AirCore sampler for tripling sampling frequency during in-situ measurements of $\delta^{13}CH_4$ (Defratyka et al., 2021). The LSCE instrument was calibrated using a 3-point mole fraction and isotopic composition calibration, just before instrument's shipment to the UK. After calibration, $CH_4$ mole fraction is reported using the WMO
X2004A scale and $\delta^{13}CH_4$ is reported using international Vienna Pee Dee Belemnite (VPDB) standard (Craig, 1957).

During mobile near-source measurements, the sampling method was based on driving through a plume of $CH_4$. At the start of the release, a vehicle intersected the plume perpendicular to the wind multiple times. Then, for the case of the RHUL mobile laboratory, bag samples were collected by pumping air into 3 litre Flexfoil bags (SKC) within the $CH_4$ plume, where the largest enhancement was observed. During the experiment, at least two bag samples from each $CH_4$ plume, plus a
background sample were collected per day. Bag samples, collected by RHUL, were measured afterward in the laboratory, using Picarro 1301 to determine $CH_4$ mole fraction and using continuous flow gas chromatography isotope ratio mass spectrometry (CF-GC-IRMS Isoprime mass spectrometer with Elementar Trace Gas module, henceforth called IRMS) to determine $\delta^{13}CH_4$ (Fisher et al., 2006).

For LSCE sampling, if during $CH_4$ plume intersection, the largest $CH_4$ enhancement achieved at least 500 nmol mol$^{-1}$ above
background, the intersected $CH_4$ plume was re-sampled using air collected and stored in the AirCore (see Appendix A). Data collected using the AirCore are henceforth called AirCore samples. During the three initial releases, bag samples were collected to be measured afterwards on the CRDS instead of on the in-situ AirCore sampler, as batteries issue occurred at LSCE mobile laboratory. Over the 24 releases, during 12 of them, AirCore sampling was performed. For most of the releases, more than one AirCore sample was collected. In total, 31 AirCore samples were collected.

Significant cross sensitivities between $C_2H_6$ and $\delta^{13}CH_4$ in the absorption spectrum can lead to bias in the measured $\delta^{13}CH_4$ by CRDS (details in Appendix A). The effect is inversely proportional to the $CH_4$ mole fraction and proportional to the $C_2H_6$ mole fraction in the sample and has been previously quantified (Rella et al. 2015; Assan et al. 2017; Defratyka 2021, chapter 2). During this study, bag samples measured by LSCE, were collected when only $CH_4$ was released ($C_2H_6:CH_4 = 0.00$), thus the $C_2H_6$ on $\delta^{13}CH_4$ correction was not applied for bag samples measured by CRDS. In the case of AirCore studies, data
treatment to determine $\delta^{13}CH_4$ source signatures is repeated twice. First, without applying the $C_2H_6$ on $\delta^{13}CH_4$ correction, and second, with the applied $C_2H_6$ on $\delta^{13}CH_4$ correction, to verify the impact of the $C_2H_6$ on $\delta^{13}CH_4$ correction for in-situ mobile measurement of $\delta^{13}CH_4$.



### 2.4. Mass conservation methods

During mobile near-source measurements, the observed $CH_4$ mole fraction and $\delta^{13}CH_4$ were a mixture of atmospheric background $CH_4$ and the $CH_4$ from the source. To determine the isotopic signature of the source, the mass conservation principle can be used. This principle is widely applied either by using the Keeling method or the Miller-Tans method (Hoheisel et al. 2019; Menoud et al. 2020; Defratyka et al. 2021; Fernandez et al. 2022). In the Keeling method (Keeling, 1961; Pataki et al., 2003), $\delta^{13}CH_4$ is plotted against the inverse of $CH_4$ mole fraction and the y-intercept of the fitted linear regression is interpreted as the $\delta^{13}CH_4$ of the observed source:

$$\delta^{13}CH_{4\,obs} = \frac{CH_{4\,bckg}}{CH_{4\,obs}} \cdot \left( \delta^{13}CH_{4\,bckg} - \delta^{13}CH_{4\,s} \right) + \delta^{13}CH_{4\,s} \quad (1),$$

where subscripts obs, bckg and s refer to observed, background and source values.

The Miller-Tans method (Miller and Tans, 2003) is another mass conservation approach, where the mole fraction and the isotopic signature of atmospheric background are assumed to be well known. The isotopic signature of the source is represented by the slope of a fitted linear regression, where, after background subtraction, $\delta^{13}CH_4$ multiplied by $CH_4$ mole fraction is plotted against $CH_4$ mole fraction:

$$\delta^{13}CH_{4\,obs} \cdot CH_{4\,obs} - \delta^{13}CH_{4\,bckg} \cdot CH_{4\,bckg} = \delta^{13}CH_{4\,s} \cdot \left( CH_{4\,obs} - CH_{4\,bckg} \right) \quad (2).$$

The Miller-Tans method can be useful to interpret studies, where the Keeling method assumption of stable background is unfulfilled or unknown, e.g. when studies are conducted over a long period of time (Lowry et al., 2020; Al-Shalan et al., 2022).

### 3. Analytical methods of the acquired measurements

Statistical properties of the $\delta^{13}CH_4$ source signatures determined, with methods detailed in section 2, can be verified with a few steps to find the best analytical strategy for signature determination (Fig. 1). For this purpose, data collected using different mobile sampling strategies (bag samples measured on IRMS, bag samples measured on CRDS and CRDS AirCore in situ sampling) are analysed, both using Keeling method and Miller-Tans method, while different backgrounds (Sect. 3.1), linear fitting methods (Sect. 3.2) and averaging strategies (Sect. 3.3) are employed.

### 3.1. Background determination for Miller-Tans method

To evaluate the impact of a chosen background $CH_4$ mole fraction and $\delta^{13}CH_4$ signature, differently defined backgrounds are subtracted for Miller-Tans method. For bag samples measured on IRMS, as a first attempt, an "individual background" was subtracted, defined as a background bag sample collected directly after the release, when bag samples were collected within $CH_4$ enhancement. For example, for all bag samples collected during first day, the background sample collected on the first day was subtracted. For the next calculation, an "averaged background" is subtracted, which is defined as the average of all background bag samples collected over whole experiment. Next, to verify the sensitivity of Miller-Tans method for a



differently defined background, calculations for two backgrounds with lower $CH_4$ mole fraction and $\delta^{13}CH_4$ than during the experiment were conducted: "global" and "random" background. A global background is an average global $CH_4$ mole
fraction observed in September 2019, equals to $1.8707 \pm 0.0011$ µmol mol$^{-1}$ (NOAA/ESRL). As $\delta^{13}CH_4$ observed at Mace Head in September 2019 was equal to -47.9 ‰, what was similar to background $\delta^{13}CH_4$ measured during controlled experiment, global $\delta^{13}CH_4$ was defined using value from Brownlow et al., as $-47.2 \pm 0.2$ ‰ (2017). For random background, the $CH_4$ mole fraction is set up the same as the global background, but the $\delta^{13}CH_4$ was randomly set to $-42.7 \pm 0.2$ ‰ to significantly differ from other $\delta^{13}CH_4$ backgrounds to better test the sensitivity of Miller-Tans method to subtracted
background.

For bag samples measured on CRDS, Miller-Tans method is implemented three times, while differently defined backgrounds are subtracted. The backgrounds have been chosen similarly as for IRMS analysis. Thus, analysis is implemented three times where individual, averaged, and global background is subtracted. Background $CH_4$ mole fraction and $\delta^{13}CH_4$ for bag samples measured on IRMS and CRDS are presented in Appendix A.

In the case of in-situ AirCore sampling, for the Miller-Tans method, data were analysed twice. First, subtracted background is calculated individually for each AirCore, as an average of AirCore data of an individual AirCore sample, observed directly before and after $CH_4$ elevation (Miller-Tans 1). Second, averaged background of bag samples measured on CRDS is subtracted for every AirCore sample (Miller-Tans 2).

### 3.2. Linear Fitting method

Both Keeling and Miller-Tans methods rely on linear regression fitting. Thus, to quantify the impact of the fitting method, we apply the different methods to the varying datasets across sampling techniques (Fig. 1). Our analysis is focused on methods which were used in the past to determine $\delta^{13}CH_4$ from near-source mobile measurement campaigns: Ordinary Least Squares (OLS) (Defratyka et al. 2021), Major Axis (MA) (Menoud et al., 2022), York fitting (Hoheisel et al., 2019) and Bivariate Correlated Errors and Intrinsic Scatter (BCES) Orthogonal (e.g. Fernandez et al. 2022). The MA method is also
known as Orthogonal Distance Regression (ODR) or Deming regression. Most of the tested fitting are calculated using built in packages and functions in R: OLS using lm() function, OLS II and MA using lmodel2() function and York fitting using York() function from package IsoplotR. As there is no available package to calculate BCES fitting in R, BCES fitting is calculated using the lnr module in python.

OLS method minimizes distance only on y-axis between fitted line and data points, using the principle of least squares to
minimise the sum of the vertical distances from the regression line, what is also known as model I regression method (Legendre and Legendre 1998, chapter 10). In the presence of measurement errors in both x and y variables, the OLS method can be only used if the x value is measured with little error, compared to the y value error. According to Legendre and Legendre (1998), if the error rate on y axis is more than three times than on x, OLS is the most efficient method to estimate slope of linear fitting. Thus, using lmodel2() function, OLS can be also calculated, what was done here (further OLS II). It



allows for comparison of OLS results obtained by two different functions (lm() and lmodel2()), which supposes to give the same results for OLS and OLS II methods.

If both x and y variables are not controlled by the researcher or measured with an error, using OLS can cause an underestimation of the slope inferred by the linear regression (Legendre and Legendre 1998, chapter 10). Thus, the model II linear regression methods are recommended because they minimize the distance both of x and y from the regression line. MA method minimize the sum of the squared Euclidean distances (x and y distances) from the regression line and it is examined here as an example of model II linear regression methods. Geometric mean regression (GMR) is another model II linear regression method, but is not tested in this study as it is expected to deliver similar results to the MA method (Zobitz et al., 2006). Details about standard errors of OLS and MA methods are presented in Appendix A.

In contrast to OLS and MA methods, York fitting (York et al., 2004) and BCES regression (Akritas and Bershady, 1993) allow inclusion of x and y errors. Overall, York fitting can be treated as a general linear regression method, while OLS and MA are special cases valid in particular conditions and can be obtained mathematically from York fitting when appropriate circumstances appears (York, 1966; York et al., 2004). In the York fitting method, the best slope fit is searched iteratively, where the initial slope value is assumed, e.g. using OLS. Then, computations are weighted, based on x and y measurement errors. Finally, computations are repeated until differences between iteration are smaller than tolerance level, e.g. $10^{-15}$ (York et al., 2004).

BCES method is a direct extension of OLS and was a last verified linear fitting method. Within BCES, four sub-methods can be employed: BCES (Y|X), BCES (X|Y) and two symmetric lines: BCES bisector and BCES Orthogonal (Akritas and Bershady, 1993). BCES (Y|X) assume x as the independent variable. BCES bisector was shown to be self-inconsistent and should not be used (Hogg et al., 2010). Finally, BCES Orthogonal is a line which minimizes orthogonal distances and should be particularly used when it is not clear which variable is supposed to be treated as the independent value. Our study is focused on the application of BCES Orthogonal, as this method was broadly implemented in previous studies (e.g. Zazzeri et al. 2015; Lowry et al. 2020; Fernandez et al. 2022). Additionally, to examine the difference between two BCES methods, BCES (Y|X) is also tested, as both methods could be implemented to determine $\delta^{13}CH_4$.

To arrive to the final uncertainty of x- and y-axis, error propagation was applied, both for Keeling and Miller-Tans methods. Details of used error propagation are presented in Appendix A.

### 3.3. Data averaging

#### 3.3.1. Data averaging bag samples measured by IRMS and CRDS

In the long-term perspective, on some sites multiple visits are made over a few years to collected bag samples (e.g. Lowry et al. 2020). To report $\delta^{13}CH_4$ source signature from multiple visits, determined $\delta^{13}CH_4$ are averaged. Thus, in this study we verify the impact of the chosen averaging strategy on averaged $\delta^{13}CH_4$. In a "treatment 1" averaging approach, $\delta^{13}CH_4$ is calculated separately for each individual day and the final $\delta^{13}CH_4$ is calculated as an average of determined $\delta^{13}CH_4$ source




signatures for individual days. In a "treatment 2" averaging approach, the bag samples results are treated as one data set and $\delta^{13}CH_4$ of methane source with its uncertainty is determined directly from the linear regression.

### 3.3.2. Data averaging AirCore in-situ sampling

For AirCore in-situ sampling, the observed $\delta^{13}CH_4$ is still noisy and their fluctuation can have a potential impact in determining $\delta^{13}CH_4$ source signatures using mass conservation methods. To check if data smoothing improves determined $\delta^{13}CH_4$, data are cumulated and averaged in clusters before being analysed. In total, 6 data sets have been prepared from each AirCore sample and are analysed using mass conversion methods: raw data, three clusters based on $CH_4$ mole fraction bins, with steps of 10 nmol mol$^{-1}$, 50 nmol mol$^{-1}$ and 100 nmol mol$^{-1}$ and two time average clusters with 10 s and 15 s time

averaging steps (Fig. 1). Examined clusters are chosen arbitrarily as a compromise between smoothing and potential bias due to over-averaging.

Typically, individual AirCore samples contains between 50-80 measurement points, where both $CH_4$ mole fractions and $\delta^{13}CH_4$ change over time. Similar to Hoheisel et al. 2019, AirCore sample measurement errors of individual data points are linearly interpolated based on laboratory tests (details in Appendix A). The interpolated uncertainty of individual points is

used as the uncertainty for the clusters of raw data, for both $CH_4$ mole fraction and $\delta^{13}CH_4$. However, when data points are clustered based on $CH_4$ bins or time averaging, a total uncertainty of clustered data points are a combination of both the uncertainty of clustering and clustered individual points (details in Appendix A). Interpolated uncertainties for raw data and total uncertainty for clustered data are used for York fitting and BCES regression as uncertainty of individual AirCore samples.

**3.4. Rejection criteria for AirCore samples**

After determination of $\delta^{13}CH_4$ and its uncertainty, rejection criteria are applied to every AirCore sample, to select which result should be kept for further analysis and comparison. For all mass conservation method, determined $\delta^{13}CH_4$ is rejected if the standard error of the fitted regression line is bigger than empirically chosen 5 ‰, based on Picarro CRDS performance. Based on previous studies (Defratyka 2021), an additional criterium, based on the value of $r^2$ parameter was also applied for

Miller-Tans method and the results are rejected also if $r^2 < 0.85$ to achieve a good quality of the retained $\delta^{13}CH_4$ values. This additional criterium was not previously applied for Keeling method $CH_4$ measurements, so here we look closer at the variance of $r^2$ via Keeling method to examine if $r^2$ criterium can be applied for Keeling method analysis.

Eventually, all non-rejected AirCore $\delta^{13}CH_4$, from one analytical strategy (cluster, mass conservation approach, fitting method) are averaged as a final $\delta^{13}CH_4$ for an individual strategy and are used to compare results from different analytical

approaches (Fig. 1).





### 3.5. Analytical methods recapitulation

Figure 1 presents the steps to analyse statistical properties of determined $\delta^{13}CH_4$ of methane source. For bag samples measured on IRMS and CRDS, determination of $\delta^{13}CH_4$ using six regression methods (OLS, OLS II, MA, York, BCES (Y|X) and BCES Orthogonal) and treatment 1 and treatment 2 averaging approach was implemented, both using Keeling
method and Miller-Tans method. For Miller-Tans method, calculations are repeated using different backgrounds.
For each AirCore sample, 6 differently clustered datasets were analysed using Keeling and Miller-Tans methods. For Miller-Tans method, two different backgrounds were subtracted: individual AirCore background and averaged bag samples background. The analysis is repeated using different regression methods: OLS, OLS II, MA, York, BCES (Y|X) and BCES Orthogonal (Fig. 1).
The main objective of this ensemble of analyses is to find the best strategy to determine $\delta^{13}CH_4$ of a methane source from near-source mobile measurements. As we tested numerous techniques (Fig. 1), for clarification and simplicity, we present only the most meaningful results in the result section. A more exhaustive analysis is presented in Appendixes B and C.

## 4. Results

### 4.1. Bag samples measured on IRMS and CRDS

After rejection of the 11 µmol mol-1 bag sample, which biased IRMS results (see Appendix B), IRMS data from 21 bag samples were analysed using different mass conservation methods (Table 1). CH4 mole fraction in remaining samples varied between 1.94 µmol mol-1 and 7.52 µmol mol-1. For Keeling method, differences between determined δ13CH4 using different fitting methods are statistically insignificant. The largest uncertainty is observed for OLS II and MA for treatment 1, where uncertainty is calculated from 95% confidence intervals. The smallest uncertainty is observed for York fitting for
both averaging approaches.
In the next step, IRMS data are analysed using the Miller-Tans method while different backgrounds are subtracted (Table 1). In the case of subtracting an individual background, the results of averaging treatment 1 method gives the same results as Keeling method, while the results of averaging treatment 2 are about 0.20 ‰ enriched (but York fitting), however still within 1SD agreement for all fitting methods. As with the Keeling method, for Miller-Tans with subtracted individual backgrounds,
the smallest discrepancy between treatment 1 and 2 is observed for York fitting. Afterward, IRMS data are further assessed using Miller-Tans analysis, where three different backgrounds are subtracted: averaged, global, and random. Overall, no significant differences between the results of Miller-Tans with different backgrounds subtracted are observed (Table 1).
Afterward, bag samples measured using CRDS were analysed (Table 2) with Keeling and Miller-Tans methods with three different backgrounds subtracted: individual, averaged, and global. Overall, except for the BCES Orthogonal method, all
CRDS results were more depleted, about ~0.7 ‰ or more than IRMS results. Also, as IRMS δ13CH4 precision is better than CRDS instrumentation, uncertainty of determined δ13CH4 is larger for CRDS result (Fig. 2). Comparing Keeling and



Miller-Tans methods with different subtraction backgrounds, both treatment 1 and 2 results are in good agreement between each other, despite MA fitting for Keeling method and BCES Orthogonal for all mass conservation methods.

Additionally, for both IRMS and CRDS, BCES (Y|X) and BCES Orthogonal are compared. For IRMS, depending on analytical strategy, no difference or slight difference in determined δ13CH4 are observed (Table 1). A different situation is observed for CRDS data, possibly due to the significant uncertainty of CRDS data points. While BCES (Y|X) is in good agreement with other linear fittings, results from BCES Orthogonal are biased significantly toward more depleted or more enriched values, depending on analytical strategy (Table 2). Possibly, observed bias using BCES Orthogonal is caused by forces symmetry implemented in this fitting method. As a conclusion, BCES Orthogonal should not be used for CRDS data.

Finally, comparing results from bag samples measured on IRMS and CRDS, it is clearly visible that uncertainties of CRDS results are higher than of IRMS, due to the lower precision of the instrument (Fig. 2). Additionally, δ13CH4 determined using CRDS is more depleted, about ~0.7‰, compared to IRMS results. As CRDS instrument was calibrated before the experiment, observed difference is related to the CRDS performance during bag samples measurement. Note that treatment 2 introduces some bias toward more enriched values for Miller-Tans methods, thus this averaging method should not be used in the future.

### 4.2. In-situ CRDS AirCore measurements

As well as for bag sampling, data from in-situ measurements using CRDS with the AirCore are analysed to verify the impact of different analysis strategies used for larger data sets with lower precision than for IRMS studies. In total, 31 AirCore samples were collected, but two of them were rejected for further analysis, due to CRDS cavity pressure and temperature instability during specific measurements (Appendix B). Here, we analysed the data using both the Keeling and Miller-Tans methods (Table 3), following steps presented in Fig. 1. To determine the best analytical strategy for AirCore studies, $\delta^{13}CH_4$ from IRMS bag samples equal to -40.25 ± 0.09 ‰ were treated as a reference value.

First, the $C_2H_6$ on $\delta^{13}CH_4$ correction was not applied. Overall, including data from all measurement days, for most analytical strategies, the determined $\delta^{13}CH_4$ was more depleted from AirCore studies than from IRMS, while observed bias depended on chosen strategy. As expected, due to the lower precision of CRDS than IRMS, the uncertainty of determined $\delta^{13}CH_4$ was higher than for IRMS bag samples.

Considering raw data clustering (Table 3, Fig. 3), for OLS, OLS II, York and BCES (Y|X) the observed $\delta^{13}CH_4$ was about 1 ‰ depleted compared to the IRMS results, and slight differences were observed between the Keeling and the two Miller-Tans methods. However, for these fitting methods, observed differences were statistically irrelevant and the results were in good agreement within each other. Similar to bag samples measured on CRDS, larger and significant discrepancies were observed using MA and BCES Orthogonal methods. Notable, only for BCES Orthogonal fitting, results from Miller-Tans 1 and Miller-Tans 2 were significantly different, which appears unrealistic. Regarding observed biases, MA and BCES Orthogonal should not be used to analyse CRDS AirCore data. These methods force symmetry between x- and y- axis,



which causes bias in the determined slope and intercept of the fitted line, as y-axis values are less precise and vary more than
on the x-axis.

Subsequently, the impact of clustering data on the final averaged $\delta^{13}CH_4$ was tested (see Appendix C). Overall, our study shows that averaging causes a changeable bias, which depends on the clustering method and the linear fitting. Additionally, clustering increases the uncertainty of the final, averaged $\delta^{13}CH_4$. Furthermore, depending on the clustering method and the linear fitting used, the amount of rejected individual AirCore samples varies. The largest discrepancies between raw and clustered data are observed for the MA and BCES Orthogonal linear fitting methods. As clustering has a negative impact for the results, our recommendation here is to only use raw data for further analysis.

Based on previous experience (Defratyka et al. 2021), for Miller-Tans method, individual AirCore sample results are rejected if their uncertainty is greater than 5 ‰, and if $r^2$ is less than 0.85, in order to balance precise results and the quality of the retained values. Here, for Keeling method, only criterium of uncertainty lower than 5‰ is applied, and an attempt has been made to find the best $r^2$ value, below which AirCore results should be rejected. However, for CRDS AirCore studies, the $r^2$ values remain low, mostly ranging between 0.1 and 0.3, with no visible trend of increasing $r^2$ values as the Keeling method results approach IRMS bag samples results. Thus, due to low $r^2$ values, it was not possible to find a satisfying $r^2$ rejection criterium, which could possibly introduce some bias using the Keeling method to CRDS AirCore results. Additionally, as the only uncertainty criterium is applied to Keeling method results, $\delta^{13}CH_4$ of individual AirCore samples is more spread (Fig. 3), which increases the uncertainty of the final, averaged $\delta^{13}CH_4$. Thus, we recommend using the Miller-Tans method instead of the Keeling method mass conservation approach to determine $\delta^{13}CH_4$ while using CRDS with an AirCore.

Afterward, all analyses were repeated when $C_2H_6$ on $\delta^{13}CH_4$ correction is applied (Fig. 4). By applying a $C_2H_6$ correction, for all analytical strategies, the final averaged $\delta^{13}CH_4$ is shifted towards more carbon 13 depleted values compared to uncorrected data. For raw data (Fig. 4, Appendix C), this bias toward negative values reaches ~2 ‰ or more, depending on the type of linear regression. Therefore, the $C_2H_6$ on $\delta^{13}CH_4$ correction introduces additional bias, resulting in the final averaged $\delta^{13}CH_4$ to be more biased compared to the IRMS reference value. This leads us to recommend refraining from using $C_2H_6$ on $\delta^{13}CH_4$ corrections for CRDS AirCore measurements, even in the presence of $C_2H_6$ in the AirCore sample. The negative impact of $C_2H_6$ on $\delta^{13}CH_4$ corrections can come from the method to determine the correction, which includes correction due to cross sensitives of $C_2H_6$ with $H_2O$, $CH_4$ and $CO_2$. Notably, $H_2O$ has the biggest impact for $C_2H_6$ reported by CRDS G2201-i. Possibly, in the case of sampling dried air, $C_2H_6$ has neglected impact on $\delta^{13}CH_4$, thus using $C_2H_6$ on $\delta^{13}CH_4$ correction biased data, which initially do not require $C_2H_6$ correction.

Finally, we observed that individual AirCore values for samples collected on days 4 and 5 of the controlled release experiment are more depleted than samples collected in days 2 and 3 (Fig. 3). It is possible, that an unnoticed problem occurred with the instrument calibration or encountered mobile set-up leaks during those days. Based on this, we recommend measuring the calibration gases on each measurement day, both before and after the fieldwork. Due to observed shifts during the last two days, the final calculated averaged $\delta^{13}CH_4$ only included days 2 and 3 measurements (Table 4). As a result, for



uncorrected data and using Miller-Tans method with OLS, OLS II, York and BCES (Y|X) linear regressions, the difference between the IRMS reference and AirCore $\delta^{13}CH_4$ values are statistically non-significant (Fig. 4).

### 4.3. Direct $\delta^{13}CH_4$ measurements

As a final step of data analysis, the sample taken directly from the cylinder was compared with indirect, near-source IRMS and CRDS measurements. $\delta^{13}CH_4$ measured by laser spectrometry is equal to -41.45 ± 0.06 ‰ (1SD), while $\delta^{13}CH_4$ measured by IRMS achieved -41.27 ± 0.06 ‰ (1SD) (Rennick et al., 2021). The value difference between these two instruments is equal to 0.18 ‰. Discrepancy between laser spectrometry and IRMS can be ignored, according to Umezawa et al. (2018) the variability between different IRMS instruments can be up to 0.5‰, depending on the calibration, correction strategy, and type of the instrument.

Compared to our indirect, near-source measurements, direct $\delta^{13}CH_4$ measurements resulted in more depleted values. For IRMS bag samples, the discrepancy between direct and indirect studies achieves ~ 1 ‰. As the uncertainty of both methods are small (1SD = 0.06 ‰ for direct studies and uncertainty for York fit = 0.014 ‰ for indirect studies), such observed discrepancies for direct and indirect measurements of $\delta^{13}CH_4$ are significant. The averaged CRDS AirCore $\delta^{13}CH_4$ from days 2 and 3, shows a similar discrepancy to the direct studies as observed for the IRMS bag samples. However, uncertainties of CRDS AirCore results are much larger than for IRMS results (2.62 ‰ for York fitting). CRDS bag samples are more $^{13}C$ depleted than other indirect methods (-41.02 ± 6.68 ‰), making these indirect measurements compatible with the direct ones (because of larger errors).

Notably, direct and indirect samples were collected in different conditions. For indirect studies, the gas was released 45 minutes from cylinder at high speeds (up to 70 l min$^{-1}$), what was causing cooling of the released gas. For direct sampling, gas was released from one cylinder to another in less than two minutes, thus the change of the temperature was negligible. Potentially, these two different sampling collection approaches could cause different fractionation effects, which would explain the observed discrepancies. Since all releases had release speed between 35 and 70 l min$^{-1}$, it was not possible to compare the impact of high and low speeds. As this observed discrepancy was not expected before the experiment, the potential impact of released gas temperature was not tested in this study. Further studies on possible isotopic fractionation during gas release are planned in the future to verify this hypothesis.

## 5. Discussion

### 5.1. Recommendation for the best analytical strategy

Our study aims to find a unified analytical strategy for determining $\delta^{13}CH_4$ source signatures, eliminating the need to choose between biased methods or switch between methods depending on the conditions. With the increasing popularity of CRDS instruments for measuring source signatures, it is crucial to evaluate the performance of both IRMS and CRDS in determining $\delta^{13}CH_4$. The novelty of the study is the comprehensive inter-comparison between (i) indirect studies of $\delta^{13}CH_4$





using bag sampling measured afterwards both by IRMS and CRDS, (ii) in-situ CRDS with an AirCore storage tool and (iii)

direct measurements from gas cylinders. We observe that due to high precision and accuracy of IRMS instruments, the
chosen mass balance approach and linear fitting method do not significantly affect IRMS results. However, as CRDS
instrument is less precise, more precaution should be taken to assure robust reporting of $\delta^{13}CH_4$ measurements.

Overall, due to the observed bias compared to IRMS results and higher uncertainty, we do not recommend measuring bag
samples using CRDS. Thus, we strongly recommend using only IRMS for analysing bag samples. To analyse IRMS data,

both Keeling and Miller-Tans methods can be used. However, in the case of the Miller-Tans method, individual background
should be subtracted. Bag samples collected during different days should not be treated as one dataset. Instead, $\delta^{13}CH_4$
should be calculated for individual days and then averaged. We have found that $\delta^{13}CH_4$ determined using in-situ CRDS
AirCore measurements agrees well with the IRMS results. For CRDS AirCore studies, we recommend using the Miller-Tans
method, with the subtraction of the individual background. To obtain robust and accurate results, raw, non-clustered data

should be analysed. As $C_2H_6$ on $\delta^{13}CH_4$ correction introduces bias compared to the IRMS results, we do not recommend
using the correction developed for CRDS during AirCore studies. For consistency, we recommend using either York or
BCES (Y|X) fitting methods for both IRMS bag samples and CRDS AirCore, as they include the uncertainty of measurement
points and give the most consistent results. The OLS method can also be applied to determine $\delta^{13}CH_4$, as differences
between York, BCES (Y|X) and OLS fitting methods are statistically irrelevant. However, in the case of a lower $CH_4$ range

or higher uncertainty of measured $\delta^{13}CH_4$, the discrepancy between York and OLS methods can increase. For CRDS AirCore
studies, we strongly discourage the use of the MA and BCES Orthogonal methods as their forced symmetry introduces
varying biases. Following these recommendations will decrease the risk of obtaining inaccurate and imprecise $\delta^{13}CH_4$ source
signatures.

### 5.2. Comparison with previous studies

Few studies have been conducted to find the best strategy for applying Keeling or Miller-Tans methods to determine isotopic
signatures, and they focused on continuous measurements of $CO_2$ (Pataki et al., 2003; Miller and Tans, 2003; Zobitz et al.,
2006; Wehr and Saleska, 2017). Pataki et al. (2003) concentrated on the application of Keeling method for $\delta^{13}C$ of $CO_2$.
However, as they highlighted in their paper, this method can be used also for methane and other isotopic ratios, where each
application has its own constraints. Pataki et al. (2003), and Miller and Tans (2003) recommend using the model II (e.g. MA)

fitting method for mass conservation because the OLS method could introduce a systematically bias, especially if the linear
fitting $r^2$ value is low. However, Zobitz et al. (2006) showed that model II can also introduce some bias, especially if the
range of the $CO_2$ mole fraction is low (e.g. $CO_2$ enhancement above background is lower than 20 µmol mol$^{-1}$) and if
variability on the x-axis is much lower than in y-axis. Geometric mean regression (GMR) is another model II linear
regression method that has not been tested in our study, as it is expected to yield similar results to the MA method (Zobitz et

al., 2006). In our study, we did observe bias for the MA method for CRDS studies, where uncertainty and fluctuation of the





measured $\delta^{13}CH_4$ is greater than for $CH_4$ mole fractions. Here, measured $CH_4$ mole fraction exceeds by at least 0.5 µmol mol$^{-1}$ of the background mole fraction. Thus, providing a signal-to-noise ratio which was large enough to not introduce biases in the case of high precision IRMS measurements using model II method. However, bias due to low signal-to-noise ratio can occur when observing lower enhancements. For example, this is typically the case for measurement stations located 415 at some distance from the source conducting continuous measurements.

Finally, Wehr and Saleska (2017) proposed using York fitting to determine $\delta^{13}CH_4$, as it is the most general regression method, which also accounts for uncertainties of both the x- and y-axis. Based on Monte Carlo simulations, used to determine the isotopic signatures of $CO_2$, they presented that York fitting produces the closest reality results, compare to OLS and GMR methods. Their conclusion aligns with our study, as the York fitting method consistently provides robust 420 results for all examined analytical approaches. Additionally, we observe smaller discrepancies between OLS and York fitting methods compared to the studies of Wehr and Saleska (2017). This can be explained by the larger $CH_4$ enhancements relative to $CO_2$ enhancements experienced in our study compared to theirs.

### 5.3. Possible improvements and further applications

Based on our study, there are several analytical details worth special attention during measurements of $\delta^{13}CH_4$. First, the 425 CRDS instrument was used in the $CO_2$-$CH_4$ simultaneous mode. According to the manufacturer, conducting measurements in $CH_4$ isotope-only mode would increase instrument precision and frequency, therefore improving results of CRDS measurements. Furthermore, we observe that bag sample dilution introduced a bias for IRMS analysis, and therefore, we decided to exclude it from IRMS data. Thus, we recommend carrying out the dilution in a controlled, well-examined way to avoid introducing any fractionation.

Remarkably, we observed about a 1 ‰ discrepancy between directly and indirectly measured $\delta^{13}CH_4$. We expect that this observed discrepancy is caused by a $CH_4$ fractionation occurring due to different conditions of gas releasing during direct and indirect sampling. Thus, it is important to examine the way in which the gas is released into the atmosphere to assess whether the speed and temperature of released gas can cause any fractionation effects and potentially biases in the determined $\delta^{13}CH_4$ source signatures.

In our study, we focus entirely on finding the best analytical strategy for near-source mobile measurements to determine $\delta^{13}CH_4$ source signatures. However, we anticipate that the results can be generalize to other applications where similar isotopic mixing lines are appropriate. For example, the same conclusions should apply for the determination of $\delta D$-$CH_4$ and stable isotope ratios of $CO_2$. Also, our conclusions should be applicable for continuous isotopic measurements, both for $CO_2$ and $CH_4$. Before expanding our conclusion to other isotopes or continuous measurement studies, it is important to consider 440 that the range of observed mole fractions, signal-to-noise ratios, precision, and variability of y-axis could potentially introduce biases depending on their magnitudes and on the chosen fitting methods. Based on our study, York and BCES (Y|X) are good candidate methods to apply in different contexts, as they exhibited the least variability and incorporate uncertainties of the x- and y-axis. Furthermore, establishing rejection criteria for individual applications, such as the size of



uncertainty or the $r^2$ parameter, can identify outliers and improve the accuracy and precision of determining $\delta^{13}CH_4$ source

signatures.

## 6.    Conclusions

This study is focused on an in-depth analysis of statistical methods for the determination of $\delta^{13}CH_4$ signatures in near-source conditions. We observed good agreement between Keeling and Miller-Tans methods for IRMS bag sample measurements. We recommend using the Miller-Tans method instead of the Keeling method for CRDS AirCore studies, as the Keeling

method results indicated more bias compared to the IRMS results, chosen as a reference in this study. We do not recommend using the CRDS instrument for bag samples, as results are less precise and accurate compared to the other methods examined. We observed that MA and BCES Orthogonal methods introduce a bias to the result for CRDS data, due to forced symmetry. Thus, we recommend using the York and BCES (Y|X) linear fitting, especially as they also incorporate the uncertainty of both the x- and y- axis. We also demonstrated that OLS provides sufficiently robust results and, for simplicity,

can be used to determine $\delta^{13}CH_4$ in near-source conditions. In the case of CRDS AirCore studies, we recommend analysing raw data and refraining from applying a $C_2H_6$ correction to $\delta^{13}CH_4$, especially when sampling dry air.

The conclusions of our work provide a robust starting point for other applications that utilize isotopic mixing lines. However, the range of observed mole fractions, signal-to-noise ratios, and precision and fluctuation of isotopic signatures have the potential to introduce biases depending on their magnitude and the chosen analytical and fitting methods. Thus, as

demonstrated in our study , the applied analytical strategy must be chosen carefully.

## APPENDIX A

### Mobile laboratories used during controlled release experiment

#### RHUL mobile laboratory

RHUL's mobile kit consisted of a 4WD petrol SUV (since been replace with a hybrid Toyota RAV4 AWD) rigged out

continuous measurement instrumentation, air sample collection equipment, and a mounted mast supporting a high-precision GPS unit, 3 inlet lines (1.8 m from ground level). The GPS was connected to a Picarro CRDS G2301, measuring $CH_4$, $CO_2$, and $H_2O$ mixing ratios (~3 second frequency), equipped with a Picarro A0941 Mobile Module for matching mixing ratio measurements and GPS coordinates in real-time. This combo is powered by four 12V-110Ah batteries which last up to 9 hours. Two of the inlets connect to CRDS instruments, the Picarro and a Los Gatos Research Ultra-Portable $CH_4$, $C_2H_6$

analyser (not used in this study). The third inlet is attached to a manually operated 6-12V diaphragm pump powered by a rechargeable battery for collecting air outside air into 5-3L SKC FlexFoil sample bags.



### LSCE mobile laboratory

The mobile laboratory of LSCE uses a GPS receiver Navilock NL-602U and Picarro CRDS G2201-i which measures $CO_2$, $\delta^{12}CO_2$, $CH_4$, $\delta^{13}CH_4$, and $H_2O$. The gas flow of the instrument was adjusted to ~160 sccm to ensure a faster response during mobile measurements. The instrument frequency achieved ~0.27 Hz. The instrument was calibrated using a 3-point mole fraction and isotopic composition calibration, just before instrument's shipment to the UK. After calibration, $CH_4$ mole fraction is reported using the WMO X2004A scale and $\delta^{13}CH_4$ is reported using international Vienna Pee Dee Belemnite (VPDB) standard (Craig, 1957). The measurements were made in high precision mode, and both $CH_4$ and $CO_2$ were measured ($CO_2$-$CH_4$ simultaneous mode). According to producent specification, high precision mode allows for more precise measurements of $CH_4$, than high dynamic range mode, achieving 1 standard deviation (1SD) for 30 s average equal to 5 nmol mol$^{-1}$ + 0.05% of reading $^{12}CH_4$ and 1 nmol mol$^{-1}$ + 0.05% of reading $^{13}CH_4$ (Picarro, Inc., Santa Clara, CA). Based on laboratory tests (Defratyka 2021, chapter 2), used G2201-i achieves a $\delta^{13}CH_4$ precision ~3.5 ‰ for ambient air level of $CH_4$ mole fraction. However, the precision improves up to 0.7 ‰ for $CH_4$ mole fraction about 10 µmol mol$^{-1}$.

The mobile set-up of LSCE is equipped with an AirCore sampler for higher precision during in-situ measurements of $\delta^{13}CH_4$ (Karion et al., 2010). Here, AirCore sampler consist of 50 m storage tube and valves which allow to easily switch between "monitoring" and "replay" mode (e.g. Rella et al. 2015; Defratyka et al. 2021). In monitoring mode, the car is moving and $CH_4$ elevation is observed. The air is continuously measured by the analyser and, at the same time, stored in the AirCore (Fig. A.1). Once $CH_4$ mole fraction returns to the background level, the car is stopped, and the air stored in AirCore is measured in replay mode. Based on previous studies (Rella et al. 2015; Lopez et al. 2017; Hoheisel et al. 2019; Defratyka et al. 2021), 500 nmol mol$^{-1}$ elevation above background was used as a threshold to determine if observed $CH_4$ elevation is suitable to be remeasured in replay mode. Here, for AirCore in-situ studies, measurements made in replay mode, which are analysed afterwards, correspond to tripling the sampling frequency, compared to monitoring mode. Data collected in the replay mode are further called AirCore samples.

In the case of CRDS measurements, stable cavity pressure and temperature are crucial to maintain robust measurements. To assure stability of the instrument and repeatability of the measurements, data points where cavity pressure was between 147.9 Torr and 148.1 Torr and cavity temperature between 44.994 °C and 45.006 °C were kept for further analysis. In this study, two AirCore samples did not fulfil required instrument stability and were rejected from further analysis.

### $C_2H_6$ on $\delta^{13}CH_4$ correction

Significant cross sensitivities between $C_2H_6$ and $\delta^{13}CH_4$ in the absorption spectrum cases bias in the measured isotopic signature by CRDS G2201-i. The effect is inversely proportional to the sample $CH_4$ mole fraction, as well as proportional to the $C_2H_6$ mole fraction in a sample and was already quantified in previous studies (Rella et al. 2015; Assan et al. 2017; Defratyka 2021). As presented on Fig. A.2, to apply the correction, dry air should be measured and the observed $C_2H_6$ must first be corrected from interferences from $H_2O$, $CH_4$ and $CO_2$. In the next step, corrected $C_2H_6$ values must be calibrated



against a gas standard with a known $C_2H_6$ mole fraction before applying the $C_2H_6$ on $\delta^{13}CH_4$ correction. Also, the $CH_4$ mole

fraction and $\delta^{13}CH_4$ should be calibrated before applying a $C_2H_6$ on $\delta^{13}CH_4$ correction. Determined correction values do not

change over time, thus corrections calculated in April 2019 were applied to the data from the controlled release experiment.

Based on laboratory testes made by Assan et al. (2017):

$$C_2H_{6\,corr} = C_2H_{6\,raw} + A \cdot H_2O + B \cdot CH_4 + C \cdot CO_2 \tag{A.1},$$

Where A, B, C correction parameters are taken from Assan et al. (2017), for low humidity (<0.16% of water in sampled gas)

case, for $H_2O$ (%), $CH_4$ (µmol mol$^{-1}$) and µmol mol$^{-1}$ $CO_2$ are measured by CRDS G2201-i:

A = 0.44 ± 0.03 µmol mol$^{-1}$ $C_2H_6$/% $H_2O$,

B = $8 \cdot 10^{-3} \pm 2 \cdot 10^{-3}$ µmol mol$^{-1}$ $C_2H_6$/ µmol mol$^{-1}$ $CH_4$,

C = $1 \cdot 10^{-4} \pm 1 \cdot 10^{-5}$ µmol mol$^{-1}$ $C_2H_6$/ µmol mol$^{-1}$ $CO_2$.

After the correction of $C_2H_6$ mole fractions due to interferences with $H_2O$, $CH_4$ and $CO_2$, observed by CRDS G2201-i, the

$C_2H_6$ mole fraction must be calibrated to a common scale. Finally, after calibration, $C_2H_6$ mole fractions can be used to

correct measured $\delta^{13}CH_4$. Here, after laboratory tests, the $C_2H_6$ calibration and the $C_2H_6$ correction on $\delta^{13}CH_4$ are calculated

in one step:

$$\delta^{13}CH_{4\,corr} = \delta^{13}CH_{4\,raw} - \frac{E \cdot C_2H_{6\,corr}}{CH_4} \tag{A.2},$$

Where E is equal to 24 ± 1 ‰ µmol mol$^{-1}$ $CH_4$/ µmol mol$^{-1}$ $C_2H_6$, for CRDS G22401-i used during controlled release

experiment (Defratyka 2021, chapter 2). Then, the corrected $\delta^{13}CH_4$ should be calibrated to the VPDB scale, using

calibration gases.

More details of particular corrections and calibration steps necessary to calculate $C_2H_6$ on $\delta^{13}CH_4$ corrections can be found in

Assan et al. 2017.

**Background for Miller-Tans method for bag samples**

**Applied uncertainties for OLS and MA linear fitting methods**

For OLS method, the standard error of slope and y-intercept are calculated as:

$$SE_{slope} = \sqrt{\frac{\sum(y_i - \hat{y}_i)^2}{n-2}} \cdot \frac{1}{\sqrt{\sum(x_i - x_{mean})^2}} \tag{A.3}, \qquad SE_{intercept} = \sqrt{\frac{\sum(y_i - \hat{y}_i)^2}{n-2} \cdot \left(\frac{1}{n} + \frac{x_{mean}^2}{\sum(x_i - x_{mean})^2}\right)} \tag{A.4},$$

Where:

n – total sample size,

$y_i$ – actual y axis value,

$\hat{y}_i$ – predicted from linear regression value of y axis,

$x_i$ – actual x axis value,





$x_{mean}$ – mean x axis value.

The outputs from the used lmodel2() function, implemented to calculated MA and OLS II linear fitting, include the slope and
y-intercept with their 95% confidence intervals (CI). Here, for MA method and OLS II, the standard error of slope and y-
intercept are calculated from CI, where 3.92 is a student t-factor for 95% CI and i represent slope and intercept:

$$SE_i = \left(CI_{i\ upper} - CI_{i\ lower}\right)/3.92 \tag{A5}.$$

**Uncertainty propagation for Keeling and Miller-Tans methods**

Using Keeling and Miller-Tans methods, propagation of uncertainties must be considered, as the x- and y-axis are
determined by the measured $CH_4$ mole fraction and $\delta^{13}CH_4$ which inherently vary across the range of measurements. The
error propagation of new variable f (i.e. x- and y-axis for Keeling or Miller-Tans method) is calculated using common
uncertainty propagation formula (Ku, 1966):

$$u(f) = \sqrt{\left(\frac{\partial f}{\partial x}\right) u(x)^2 + \left(\frac{\partial f}{\partial y}\right)^2 u(y)^2 + \cdots} \tag{A.6}$$

Calculated in this way uncertainties can be implemented in York fitting and BCES regression. Based on eq. (A.6) for
Keeling method:

$$x = \frac{1}{CH_4}, \quad u(x) = \frac{u(CH_4)}{(CH_4)^2} \quad (A.7), \qquad\qquad y = \delta^{13}CH_4, \quad u(y) = u(\delta^{13}CH_4) \ (A.8),$$

where:

$CH_4$ – $CH_4$ mole fraction in µmol mol$^{-1}$,

$u(CH_4)$ – measurement uncertainty of $CH_4$ in µmol mol$^{-1}$,

$\delta^{13}CH_4$ – $\delta^{13}CH_4$ isotopic signature in ‰,

$u(\delta^{13}CH_4)$ – measurement uncertainty of $\delta^{13}CH_4$ isotopic signature in ‰.

In the case of Miller-Tans method background is subtracted, both for $CH_4$ and $\delta^{13}CH_4$. The approximation that $\Delta\delta^{13}CH_4$ is
equal to $\delta^{13}CH_4$ of the sample minus background $\delta^{13}CH_4$ is used. Thus, for Miller-Tans method, propagated uncertainties of
x- and y- axis are equal:

$$x = \Delta CH_4 = CH_4 - CH_{4\ bckg} \ (A.9), \qquad\qquad u(x) = \sqrt{\left(u(CH_4)\right)^2 + \left(u\left(CH_{4\ bckg}\right)\right)^2} \ (A.10)$$

$$y = \Delta(\delta^{13}CH_4 \cdot CH_4) = \delta^{13}CH_4 \cdot CH_4 - \delta^{13}CH_{4\ bckg} \cdot CH_{4\ bckg} \ (A.11),$$

$$u(y) =$$

$$\sqrt{\left(CH_4 \cdot u(\delta^{13}CH_4)\right)^2 + \left(\delta^{13}CH_4 \cdot u(CH_4)\right)^2 + \left(CH_{4\ bcgd} \cdot u(\delta^{13}CH_{4\ bckg})\right)^2 + \left(\delta^{13}CH_{4\ bckg} \cdot u\left(CH_{4\ bckg}\right)\right)^2} \ (A.12),$$

where:





$CH_4$, $u(CH_4)$, $\delta^{13}CH_4$, $u(\delta^{13}CH_4)$ represent the same variables as for eq. A.7 and eq. A.8,

$CH_{4\,bckg}$ – subtracted background $CH_4$ mole fraction in µmol mol$^{-1}$,

$u(CH_{4\,bckg})$ – subtracted background measurement uncertainty of $CH_4$ in µmol mol$^{-1}$,

$\delta^{13}CH_{4\,bckg}$ – subtracted background $\delta^{13}CH_4$ isotopic signature in ‰,

$u(\delta^{13}CH_{4\,bckg})$ – subtracted background measurement uncertainty of $\delta^{13}CH_4$ isotopic signature in ‰.

**$\delta^{13}CH_4$ uncertainty for bag and AirCore samples**

In a "treatment 1" averaging approach, $\delta^{13}CH_4$ is calculated separately for each individual day. Then, the final $\delta^{13}CH_4$ is calculated as an average of determined $\delta^{13}CH_4$ for individual days and the final standard error of $\delta^{13}CH_4$ is calculated as:

$$u(\delta^{13}CH_4)_{treatment\,1} = \frac{\sqrt{\sum u(\delta^{13}CH_4)_{individual\,day}^2}}{\sqrt{n}} \qquad (A.13),$$

where n represents number of individual days.

Typically, an individual AirCore sample contains between 50-80 measurement points, where both $CH_4$ mole fraction and $\delta^{13}CH_4$ changes over time. Similarly to Hoheisel et al. 2019, the measurement errors of individual data points of an AirCore sample are linearly interpolated based on laboratory tests. Here, calibration standards containing 2 µmol mol$^{-1}$ (low standard) and 10 µmol mol$^{-1}$ (high standard) of $CH_4$ from natural gas were measured on 23$^{rd}$ August 2019 (Defratyka 2021, chapter 2). For $\delta^{13}CH_4$, the uncertainties measured by G2201-i achieved 3.4 ‰ and 0.7 ‰ for low and high standard, respectively. Then,

the uncertainty of individual points of an AirCore sample is calculated as the linear interpolation between 3.4 ‰ and 0.7 ‰, depending on $CH_4$ mole fraction of the individual point. The same approach was taken to determine uncertainty of individual points of an AirCore sample for $CH_4$ mole fraction.

For clustered AirCore sample data, uncertainty of clustering comes from the variability of measured individual points captured within one cluster. For $CH_4$ mole fraction, it is defined as the difference between $CH_4$ of individual points with

maximum and minimum $CH_4$ mole fraction of points gathered in one cluster. Then, the difference is divided by student t-factor for number of individual data points in the cluster to reflects impact of number of clustered points. Accordingly, $\delta^{13}CH_4$ clustering uncertainty is defined as the difference between $\delta^{13}CH_4$ of individual points with maximum and minimum $CH_4$ divided by student-factor:

$$u_{i\_clustering} = \frac{i_{max\,CH_4} - i_{min\,CH_4}}{t_n} \qquad (A.14),$$

where i stands for $CH_4$ or $\delta^{13}CH_4$ cumulated in one cluster and

$t_n$ – student t-factor for number of individual points captured in the cluster .

Then, each of the clustered points has its own uncertainty, calculated from linear interpolation. Thus, the uncertainty of clustered individual points is propagated from uncertainties of individual points. The uncertainty of clustered individual




points is calculated based on uncertainties of individual points with minimal and maximal $CH_4$ mole fraction within the

cluster, which are used to calculate uncertainty of clustering:

$$u_{i\_clustered\ individual\ points} = \sqrt{\left(u(i_{max\ CH_4})\right)^2 + \left(u(i_{min\ CH_4})\right)^2} \qquad (A.15),$$

where $u(i_{max})$ and $u(i_{min})$ come from a linear interpolation and stands for $CH_4$ or $\delta^{13}CH_4$ cumulated in one cluster.

Finally, the total uncertainty of clustered data points is an addition in quadrature of uncertainty of clustering and uncertainty of clustered individual points:

$$u_{i\_total} = \sqrt{u_{i\_clustering}^2 + u_{i\_clustered\ individual\ points}^2} \qquad (A.16).$$

In the case of clustering into mole fraction bins, some clusters contain only one data point. In this case, the uncertainty of clustered individual points is equal to uncertainty of individual data point from linear interpolation, as this situation is equivalent to cluster of raw data.

Total uncertainty for clusters with several data points (eq. A.16) or interpolated uncertainty for clusters with one data point,

are used for York fitting and BCES regression as uncertainty of individual AirCore sample. In the case of raw data, interpolated uncertainty is used for York fitting and BCES regression as uncertainty of an individual AirCore sample (Fig. A.3).

Eventually, all non-rejected AirCore $\delta^{13}CH_4$, from one analytical strategy (cluster, mass conservation approach, fitting method) are averaged as a final determined $\delta^{13}CH_4$ for an individual strategy and used to compare results from different

analytical approaches (Fig. 1). The final averaged $\delta^{13}CH_4$ of an individual analytical strategy, uncertainty $u(\delta^{13}CH_4)_{AirCore}$ is calculated as:

$$u(\delta^{13}CH_4)_{AirCore} = \frac{\sqrt{\Sigma\left(u(\delta^{13}CH_4)_{individual\ AirCore}^2\right)}}{\sqrt{n}} \qquad (A.17),$$

where n is number of averaged AirCore results for individual analytical strategy.

**APPENDIX B**

**Bag samples results**

     **Impact of 11 µmol mol-1 bag sample**

Overall, bag samples where $CH_4$ mole fractions were over 8 µmol mol$^{-1}$ must be diluted to be measured on the IRMS at RHUL due to detection limit. Potentially, the dilution could cause some fractionation effects and measured $\delta^{13}CH_4$ could be biased, while comparing to undiluted bag samples. As a linear regression is more sensitive toward extreme values, biased



maximum data point could significantly affect determined $\delta^{13}CH_4$ source signature. To verify a possible impact of dilution of bag samples above 8 µmol mol$^{-1}$ of $CH_4$ mole fraction, we compare results for dataset with and without 11 µmol mol$^{-1}$ bag sample, using 5 linear fitting methods. Overall, for each linear fitting method, the bias toward more carbon 13 enriched values is observed if 11 µmol mol$^{-1}$ bag sample is included in dataset. Note, the bias does not affect uncertainty of determined $\delta^{13}CH_4$. Obtained results show that dilutions can indeed bias calculated $\delta^{13}CH_4$. Thus, used dilution technique

should be carefully chosen to not introduce potential fractionation and bias and may be required for future verification. Based on the comparison, the bag sample with higher $CH_4$ mole fraction, equal to ~ 11 µmol mol$^{-1}$ is rejected from further analysis.

**Results for bag samples measured on IRMS and CRDS for all examined linear fitting methods**

**APPENDIX C**

**CRDS AirCore results**

**Impact of data clustering**

For the CRDS AirCore results, comparing raw and clustered data, for OLS and OLS II, results for clustered data are more depleted for clustering using $CH_4$ mole fraction (10 nmol mol$^{-1}$, 50 nmol mol$^{-1}$, 100 nmol mol$^{-1}$) and the lowest value is observed for Miller-Tans method for data clustered into 50 nmol mol$^{-1}$ bins. In the case of clusters based on time averaging

(10 s and 15 s), the difference between a reference value (IRMS bag samples, equal to -40.25 ± 0.09 ‰) and an averaged AirCore value from Miller-Tans method is slightly less for 10 s clustered data and significantly lower for 15 s clustered data. For these two clusters, Keeling method averaged results are biased toward more enriched values. Additionally, clustering data significantly increases uncertainty of the final averaged $\delta^{13}CH_4$ for Miller-Tans method. For the Keeling method, this increase is negligible. Notably, only for raw data and 10 nmol mol$^{-1}$ cluster obtained results are the same for OLS and OLS II

methods. Surprisingly, fewer individual AirCore results were rejected if data are clustered than for raw data in the case of Miller-Tans method.

Regarding the York fitting, due to clustering, more individual AirCore results are rejected than for raw data. In the case of 50 nmol mol$^{-1}$ and 100 nmol mol$^{-1}$ clusters, only one individual AirCore result remains for both clusters. Overall, for Miller-Tans method, for York fitting, due to clustering final averaged $\delta^{13}CH_4$ is more enriched than reference value and the bias

varies, depends on clustering method. For Keeling methods, bias toward negative values is observed and it also varies, depend on clustering. Also, for York fitting, the uncertainty of final, averaged $\delta^{13}CH_4$ increases using clustering.

In the case of BCES (Y|X) linear fitting, fewer individual results are rejected for clustered than for raw data, in the case of Miller-Tans method. For, both Keeling and Miller-Tans method, final, averaged $\delta^{13}CH_4$ for clustered data are more depleted than for raw data. Their uncertainties are larger than for raw data of Miller-Tans and the change is statistically irrelevant for

Keeling method.



Finally, regarding MA and BCES Orthogonal linear regression, the observed bias is much larger than for other fitting methods, more individual AirCore results are rejected applying rejection criteria and for some clustering all individual AirCore results are rejected. For MA, similarly to other fitting methods, uncertainty increases with clustering, while for BCES Orthogonal, Keeling method uncertainties decrease.

**Data availability**

The data that support the findings of this study are openly available in Defratyka, Sara (2023), "Dataset: Statistical evaluation of methane isotopic signatures determined during near-source measurements", Mendeley Data, V1, doi: 10.17632/vfbbdvp9w2.1 at https://data.mendeley.com/datasets/vfbbdvp9w2/1.

**Author contribution**

**S.M.D:** Writing – original draft, Conceptualization, Visualization, Methodology, Validation, Formal analysis, Data curation, Investigation. **J.L.F**: Conceptualization, Methodology, Investigation, Data Curation, Writing – review & editing. **R.E.F**: Conceptualization, Methodology, Validation, Resources, Writing – review & editing. **D..**: Conceptualization, Methodology, Validation, Resources, Writing – review & editing. **J.M.F:** Investigation, Data Curation, Formal analysis, Writing – review & editing. **S.B**: Investigation, Data Curation, Writing – review & editing. **C.Y.K**: Supervision, Methodology, Validation, Resources, Writing – review & editing. **J.D.P**: Supervision, Validation. **P.B**: Supervision, Conceptualization, Methodology, Validation, Resources, Writing – review & editing. **T.A**: Supervision, Methodology, Validation, Writing – review & editing. **C.R**: Methodology, Validation, **J.H**: Conceptualization, Methodology, Validation, Investigation, Writing – review & editing. **N.Y**: Conceptualization, Methodology, Validation, Investigation. **E.G.N**: Writing – review & editing.

**Competing interests**

Some authors are members of the editorial board of journal Atmospheric Measurement Techniques. The peer-review process was guided by an independent editor, and the authors have also no other competing interests to declare.

**Acknowledgements**

This research has been supported by the European Union's Horizon 2020 research and innovation program (Marie Skłodowska-Curie grant no. 722479).

The controlled release experiment was part of the NERC grant New methodologies for removal of methane from the atmosphere (NE/P019641/1), which also funded the LGR UMEA instrument used in these experiments. RHUL participation in the experiment was funded by the NERC Equipt4Risk project (NE/R017360/1).




TA acknowledges the funding from The project (21GRD04 isoMET). 21GRD04 isoMET has received funding from the European Partnership on Metrology, co-financed from the European Union's Horizon Europe Research and Innovation
Programme and by the Participating States."

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





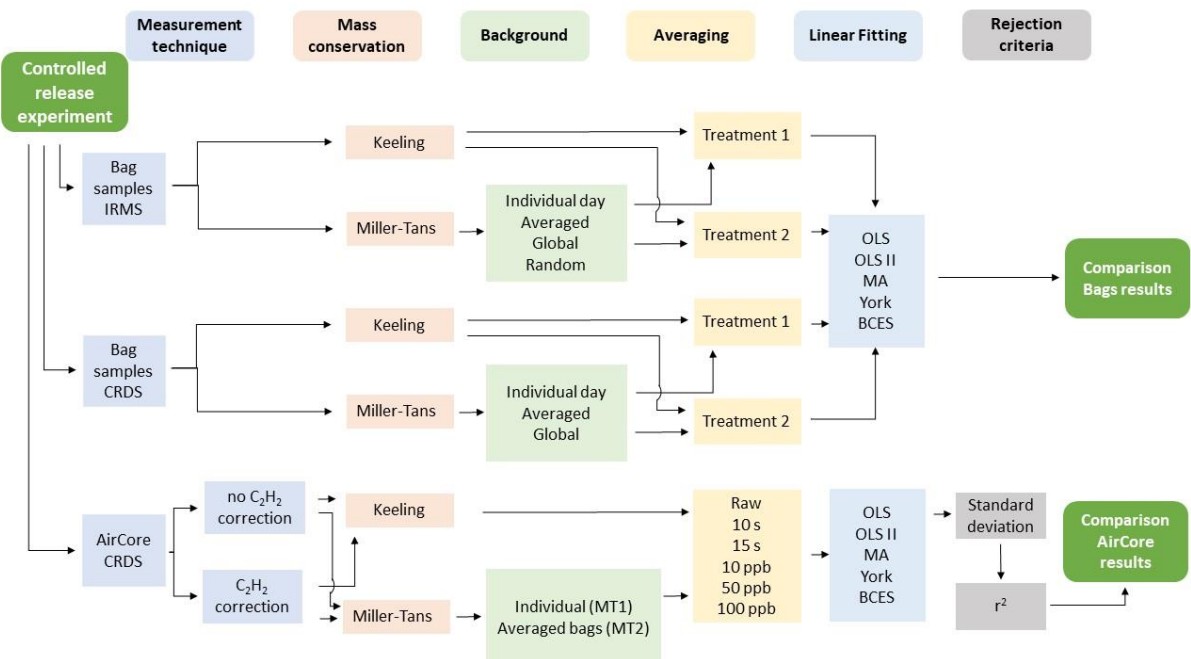

**Figure 1. Flow chart of steps to find the best analytical strategy for determination of δ¹³CH₄ source signature**

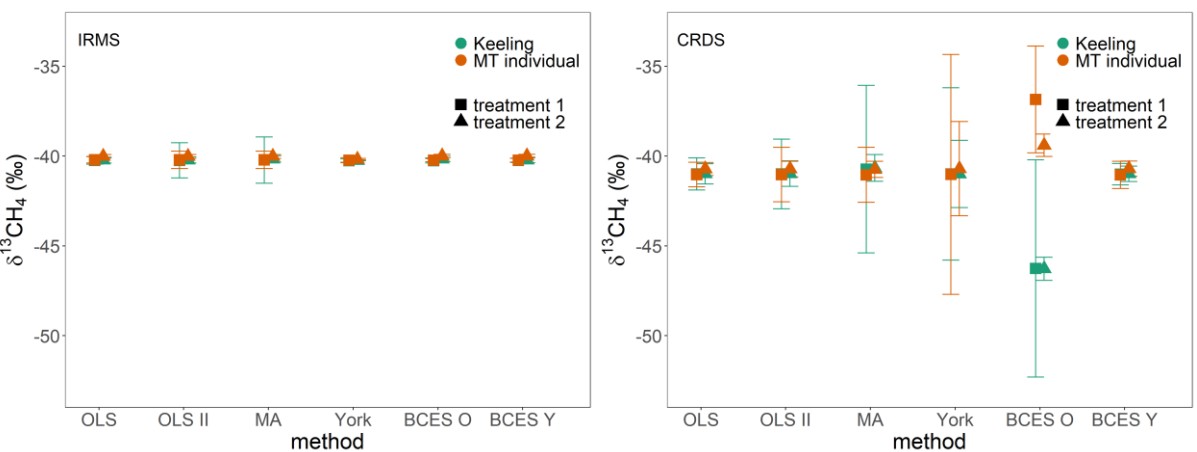

**Figure 2 Comparison of bag samples measured on IRMS (left) and CRDS (right). Keeling method and Miller-Tans method with individual background subtracted are compared. Treatment 1 and treatment 2 averaging techniques are presented.**



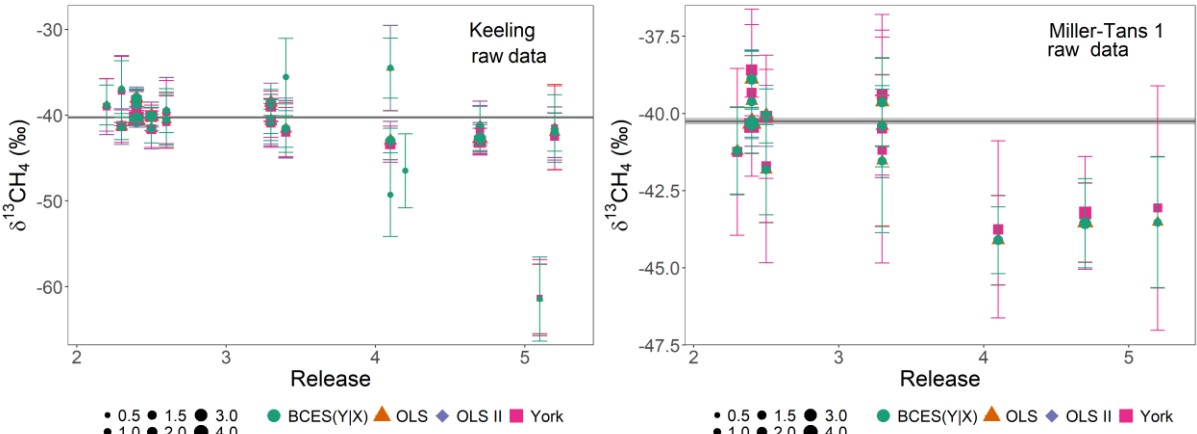

**Figure 3 Individual AirCore samples without a C2H6 on δ13CH4 correction. Size of data points corresponds to CH4 mole fraction exceed above background mole fraction in μmol mol-1. Left: Keeling method, Right: Miller-Tans method. Black line represents IRMS reference value with its uncertainty (grey line). The y-axis scale differs on left and right scale.**

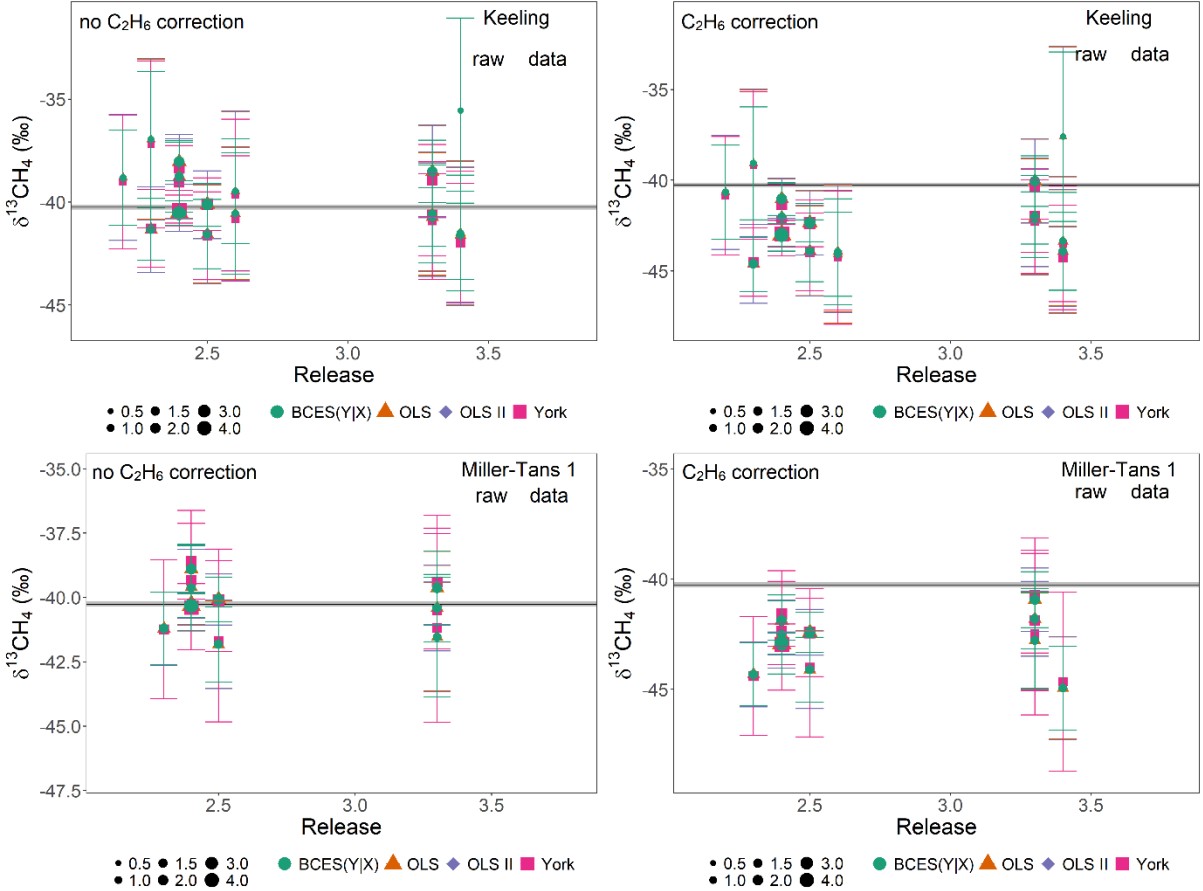

**Figure 4 Individual AirCore samples for days 2 and 3 of controlled release experiment. Points size corresponds to CH₄ mole fraction exceeding above background mole fraction in μmol mol⁻¹. Black line represents IRMS reference value with uncertainty (grey line). Left: without a C₂H₆ on δ¹³CH₄ correction. Right: C₂H₆ on δ¹³CH₄ correction applied. Top: Keeling method, bottom:**





**Miller-Tans method, individual background removed. The y -axis scale differs on left and right scale and between Keeling and Miller-Tans methods.**

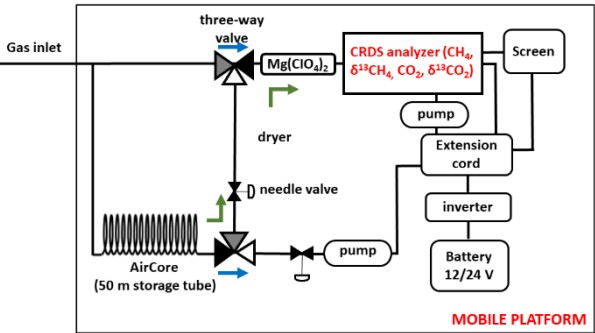

**Figure A.1. Scheme of mobile measurement set-up. The blue arrows show the airflow in monitoring mode. The green arrows show the airflow in the replay mode.**

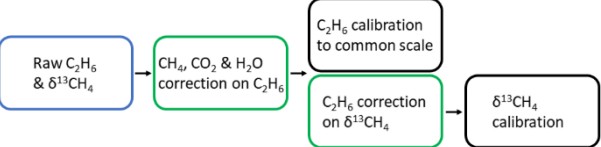

**Figure A.2. Flow chart of steps involved to determine C₂H₆ correction on δ¹³CH₄.**

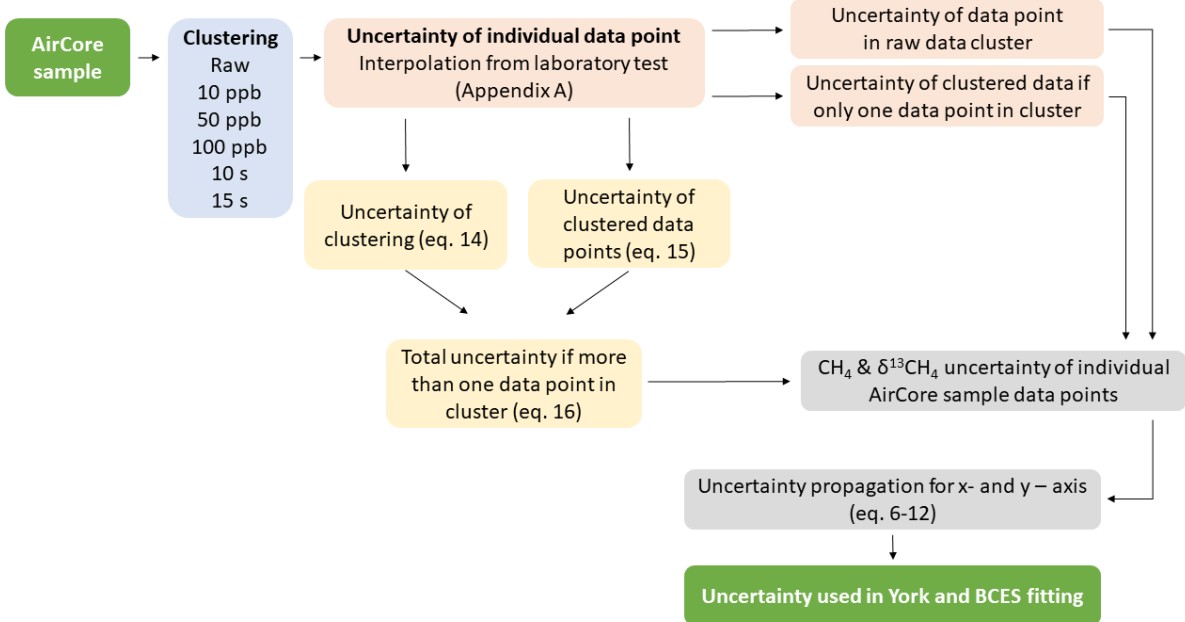

**Figure A.3. Flow chart of uncertainty calculation to use in York and BCES fitting for AirCore samples.**





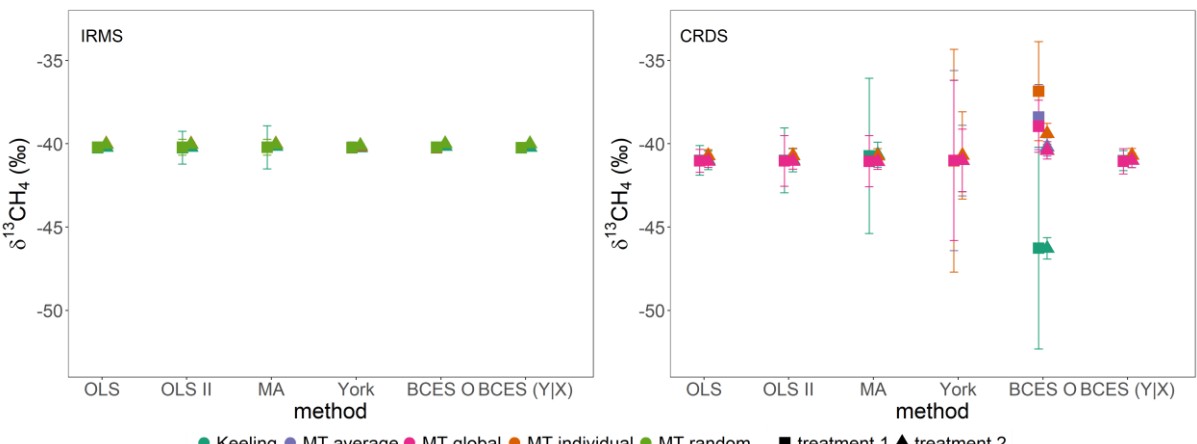

**Figure B.1 Comparison of bag samples measured on IRMS (left) and CRDS (right). Keeling method and Miller-Tans methods with different backgrounds are compared. Treatment 1 and treatment 2 averaging techniques are presented. BCES O - BCES Orthogonal, BCES Y - BCES (Y|X).**

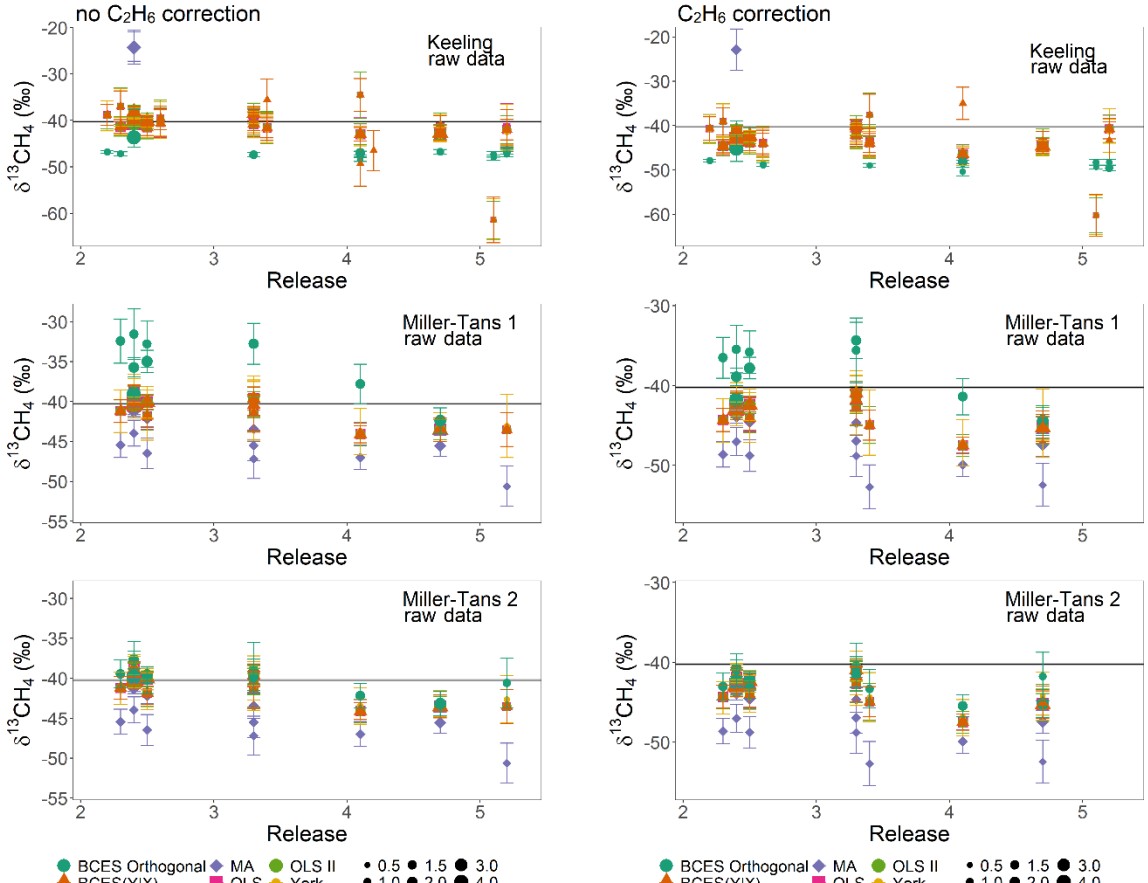

**Figure C.1 AirCore samples from Miller-Tans when individual background was subtracted. Size of data points corresponds to $CH_4$ mole fraction exceed above background mole fraction in µmol mol⁻¹. Left: $C_2H_6$ on $\delta^{13}CH_4$ correction was not applied. Right:**



C₂H₆ on δ¹³CH₄ correction was applied. The y-axis scale differs on left and right scale and between Keeling and Miller-Tans
method.

**Table 1 Keeling and Miller-Tans method results from 21 IRMS bag samples. Miller-Tans results reflect the application of 4 used subtraction backgrounds.**

| Linear Fitting | Averaging Treatment | $\delta^{13}CH_4 \pm u(\delta^{13}CH_4)$ (‰) | | | | |
| --- | --- | --- | --- | --- | --- | --- |
| | | Keeling method | Miller-Tans method individual background | Miller-Tans method averaged background | Miller-Tans method global background | Miller-Tans method random background |
| OLS | 1 | -40.24 ± 0.21 | -40.21 ± 0.17 | -40.21 ± 0.17 | -40.21 ± 0.17 | -40.21 ± 0.17 |
| OLS II | 1 | -40.24 ± 0.98 | -40.21 ± 0.48 | -40.21 ± 0.48 | -40.21 ± 0.48 | -40.21 ± 0.48 |
| MA | 1 | -40.22 ± 1.29 | -40.21 ± 0.48 | -40.21 ± 0.48 | -40.21 ± 0.48 | -40.21 ± 0.48 |
| York | 1 | -40.25 ± 0.09 | -40.23 ± 0.14 | -40.22 ± 0.33 | -40.22 ± 0.27 | -40.22 ± 0.27 |
| BCES Orthogonal | 1 | -40.25 ± 0.14 | -40.23 ± 0.09 | -40.20 ± 0.09 | -40.20 ± 0.09 | -40.20 ± 0.09 |
| BCES (Y\|X) | 1 | -40.24 ± 0.12 | -40.23 ± 0.10 | -40.23 ± 0.10 | -40.23 ± 0.10 | -40.23 ± 0.10 |
| OLS | 2 | -40.22 ± 0.16 | -40.02 ± 0.10 | -40.05 ± 0.11 | -40.05 ± 0.11 | -40.05 ± 0.11 |
| OLS II | 2 | -40.22 ± 0.17 | -40.02 ± 0.11 | -40.05 ± 0.12 | -40.05 ± 0.12 | -40.05 ± 0.12 |
| MA | 2 | -40.16 ± 0.17 | -40.02 ± 0.11 | -40.05 ± 0.12 | -40.05 ± 0.12 | -40.05 ± 0.12 |
| York | 2 | -40.24 ± 0.03 | -40.18 ± 0.05 | -40.10 ± 0.09 | -40.10 ± 0.08 | -40.10 ± 0.08 |
| BCES Orthogonal | 2 | -40.16 ± 0.15 | -40.00 ± 0.10 | -40.00 ± 0.10 | -40.00 ± 0.09 | -40.00 ± 0.09 |
| BCES (Y\|X) | 2 | -40.22 ± 0.15 | -40.00 ± 0.10 | -40.00 ± 0.09 | -40.00 ± 0.09 | -40.00 ± 0.09 |

**Table 2 Keeling and Miller-Tans method results from 8 CRDS bag samples. Miller-Tans results reflect the application of 3 used subtraction backgrounds.**

| Linear Fitting | Averaging | $\delta^{13}CH_4 \pm u(\delta^{13}CH_4)$ (‰) | | | |
| --- | --- | --- | --- | --- | --- |
| | | Keeling method | Miller-Tans method individual background | Miller-Tans method averaged background | Miller-Tans method global background |
| OLS | 1 | -41.00 ± 0.89 | -41.03 ± 0.69 | -41.03 ± 0.69 | -41.03 ± 0.69 |





| | | | | | |
|---|---|---|---|---|---|
| OLS II | 1 | -41.00 ± 1.94 | -41.03 ± 1.52 | -41.03 ± 1.52 | -41.03 ± 1.52 |
| MA | 1 | -40.73 ± 4.66 | -41.05 ± 1.53 | -41.05 ± 1.53 | -41.05 ± 1.53 |
| York | 1 | -41.00 ± 4.80 | -41.02 ± 6.68 | -41.01 ± 5.40 | -41.00 ± 4.81 |
| BCES Orthogonal | 1 | -46.26 ± 6.05 | -36.85 ± 2.97 | -38.40 ± 1.95 | -38.95 ± 1.56 |
| BCES (Y\|X) | 1 | -41.01 ± 0.60 | -41.05 ± 0.76 | -41.05 ± 0.57 | -41.05 ± 0.76 |
| OLS | 2 | -40.99 ± 0.56 | -40.72 ± 0.36 | -41.05 ± 0.38 | -41.05 ± 0.38 |
| OLS II | 2 | -40.99 ± 0.70 | -40.72 ± 0.45 | -41.05 ± 0.47 | -41.05 ± 0.47 |
| MA | 2 | -40.66 + 0.75 | -40.74 ± 0.45 | -41.07 ± 0.47 | -41.07 ± 0.47 |
| York | 2 | -41.00 ± 1.87 | -40.70 ± 2.62 | -41.01 ± 2.12 | -41.00 ± 1.88 |
| BCES Orthogonal | 2 | -46.28 ± 0.64 | -39.40 ± 0.62 | -40.20 ± 0.55 | -40.40 ± 0.52 |
| BCES (Y\|X) | 2 | -40.99 ± 0.43 | -40.70 ± 0.42 | -41.00 ± 0.44 | -41.00 ± 0.44 |

**Table 3 CRDS AirCore samples for raw cluster data. $N_{AirCore}$ represents number of AirCore samples used to determine averaged $\delta^{13}CH_4$ after applying rejection criterium. $C_2H_6$ on $\delta^{13}CH_4$ correction not applied.**

| Linear Fitting | $\delta^{13}CH_4 \pm u(\delta^{13}CH_4)$ (‰) | | | $n_{AirCore}$ | $n_{AirCore}$ | $n_{AirCore}$ |
|---|---|---|---|---|---|---|
| | Keeling method | Miller-Tans method 1 | Miller-Tans method 2 | Keeling method | Miller-Tans 1 | Miller-Tans 2 |
| OLS | -41.15 ± 3.03 | -41.22 ± 1.48 | -41.22 ± 1.48 | 22 | 12 | 12 |
| OLS II | -41.44 ± 2.93 | -41.22 ± 1.50 | -41.22 ± 1.50 | 21 | 12 | 12 |
| MA | -24.18 ± 3.38 | -44.95 ± 1.68 | -44.95 ± 1.68 | 2 | 12 | 12 |
| York | -41.67 ± 2.80 | -41.04 ± 2.72 | -40.91 ± 2.07 | 21 | 12 | 12 |
| BCES Orthogonal | -46.45 ± 1.02 | -35.51 ± 2.24 | -39.84 ± 1.89 | 19 | 9 | 12 |
| BCES (Y\|X) | -41.47 ± 2.78 | 41.23 ± 1.46 | -41.22 ± 1.46 | 25 | 12 | 12 |


**Tab A.1. Subtracted background values used for Miller-Tans method for bag samples measurements.**

| IRMS background/ CRDS background | $CH_4$ (µmol mol$^{-1}$) IRMS bag samples | $\delta^{13}CH_4$ (‰) IRMS bag samples | $CH_4$ (µmol mol$^{-1}$) CRDS bag samples | $\delta^{13}CH_4$ (‰) CRDS bag samples |
|---|---|---|---|---|
| individual day 1 | 2.0589 ± 0.0007 | -47.77 ± 0.10 | - | - |
| individual day 3 | 1.9634 ± 0.0010 | -48.12 ± 0.04 | - | - |
| individual day 4 | 1.9403 ± 0.0007 | -48.08 ± 0.06 | - | - |





| | | | | |
|---|---|---|---|---|
| individual day 5 | $1.9950 \pm 0.0012$ | $-48.30 \pm 0.02$ | - | - |
| individual release 1 | - | - | $1.9619 \pm 0.0003$ | $-47.99 \pm 3.53$ |
| individual releases 2 | - | - | $1.9810 \pm 0.0003$ | $-48.82 \pm 3.45$ |
| averaged | $1.9894 \pm 0.0009$ | $-48.07 \pm 0.23$ | $1.9715 \pm 0.0003$ | $-48.41 \pm 1.87$ |
| global | $1.8707 \pm 0.0011$ | $-47.2 \pm 0.2$ | $1.8707 \pm 0.0011$ | $-47.2 \pm 0.2$ |
| random | $1.8707 \pm 0.0011$ | $-42.7 \pm 0.10$ | - | - |

**Table B.1 IRMS bag samples results. Comparison of Keeling method and Miller -Tans individual background method with and without 11 µmol mol$^{-1}$ bag sample**

| | | $\delta^{13}CH_4 \pm u(\delta^{13}CH_4)$ (‰) | | | |
|---|---|---|---|---|---|
| Linear Fitting | Averaging | Keeling method without 11 µmol mol$^{-1}$ | Keeling method with 11 µmol mol$^{-1}$ | Miller-Tans method individual background without 11 µmol mol$^{-1}$ | Miller-Tans method individual background with 11 µmol mol$^{-1}$ |
| OLS | 1 | $-40.24 \pm 0.21$ | $-40.02 \pm 0.26$ | $-40.21 \pm 0.17$ | $-39.82 \pm 0.13$ |
| OLS II | 1 | $-40.24 \pm 0.98$ | $-40.02 \pm 1.00$ | $-40.21 \pm 0.48$ | $-39.82 \pm 0.45$ |
| MA | 1 | $-40.22 \pm 1.29$ | $-39.99 \pm 1.30$ | $-40.21 \pm 0.48$ | $-39.82 \pm 0.45$ |
| York | 1 | $-40.25 \pm 0.09$ | $-39.89 \pm 0.09$ | $-40.23 \pm 0.14$ | $-39.91 \pm 0.13$ |
| BCES Orthogonal | 1 | $-40.25 \pm 0.14$ | $-39.92 \pm 0.23$ | $-40.23 \pm 0.09$ | $-39.83 \pm 0.08$ |
| OLS | 2 | $-40.22 \pm 0.16$ | $-39.89 \pm 0.19$ | $-40.02 \pm 0.10$ | $-39.33 \pm 0.15$ |
| OLS II | 2 | $-40.22 \pm 0.17$ | $-39.89 \pm 0.20$ | $-40.02 \pm 0.11$ | $-39.33 \pm 0.16$ |
| MA | 2 | $-40.16 \pm 0.17$ | $-39.80 \pm 0.19$ | $-40.02 \pm 0.11$ | $-39.34 \pm 0.16$ |
| York | 2 | $-40.24 \pm 0.03$ | $-39.49 \pm 0.02$ | $-40.18 \pm 0.05$ | $-39.74 \pm 0.04$ |
| BCES Orthogonal | 2 | $-40.16 \pm 0.15$ | $-39.80 \pm 0.22$ | $-40.00 \pm 0.10$ | $-39.30 \pm 0.28$ |


**Table C.1 CRDS AirCore samples. $N_{AirCore}$ represents number of AirCore samples used to determine averaged $\delta^{13}CH_4$ after applying rejection criterium. $C_2H_6$ on $\delta^{13}CH_4$ correction not applied.**

| | | | $n_{AirCore}$ | $n_{AirCore}$ | $n_{AirCore}$ |
|---|---|---|---|---|---|
| Linear Fitting | Data Cluster | $\delta^{13}CH_4 \pm u(\delta^{13}CH_4)$ (‰) | Keeling method | Miller-Tans 1 | Miller-Tans 2 |



| | | Keeling method | Miller-Tans method 1 | Miller-Tans method 2 | | | |
|---|---|---|---|---|---|---|---|
| OLS | raw | -41.15 ± 3.03 | -41.22 ± 1.48 | -41.22 ± 1.48 | 22 | 12 | 12 |
| OLS II | raw | -41.44 ± 2.93 | -41.22 ± 1.50 | -41.22 ± 1.50 | 21 | 12 | 12 |
| MA | raw | -24.18 ± 3.38 | -44.95 ± 1.68 | -44.95 ± 1.68 | 2 | 12 | 12 |
| York | raw | -41.67 ± 2.80 | -41.04 ± 2.72 | -40.91 ± 2.07 | 21 | 12 | 12 |
| BCES Orthogonal | raw | -46.45 ± 1.02 | -35.51 ± 2.24 | -39.84 ± 1.89 | 19 | 9 | 12 |
| BCES (Y\|X) | raw | -41.47 ± 2.78 | 41.23 ± 1.46 | -41.22 ± 1.46 | 25 | 12 | 12 |
| OLS | 10 nmol mol$^{-1}$ | -41.47 ± 2.80 | -42.52 ± 2.22 | -42.52 ± 2.22 | 19 | 17 | 17 |
| OLS II | 10 nmol mol$^{-1}$ | -41.47 ± 2.91 | -42.52 ± 2.30 | -42.52 ± 2.30 | 19 | 17 | 17 |
| MA | 10 nmol mol$^{-1}$ | -28.55 ± 4.38 | -45.69 ± 2.57 | -45.69 ± 2.57 | 1 | 17 | 17 |
| York | 10 nmol mol$^{-1}$ | -41.86 ± 3.19 | -41.61 ± 4.13 | -41.52 ± 4.17 | 18 | 3 | 4 |
| BCES Orthogonal | 10 nmol mol$^{-1}$ | -46.50 ± 0.54 | -26.71 ± 1.91 | -7.33 ± 1.86 | 28 | 1 | 1 |
| BCES (Y\|X) | 10 nmol mol$^{-1}$ | -42.14 ± 2.70 | -42.53 ± 2.02 | -42.52 ± 2.02 | 21 | 17 | 17 |
| OLS | 50 nmol mol$^{-1}$ | -42.46 ± 2.96 | -44.05 ± 3.13 | -44.05 ± 3.13 | 17 | 19 | 19 |
| OLS II | 50 nmol mol$^{-1}$ | -42.46 ± 3.21 | -43.30 ± 3.09 | -43.30 ± 3.09 | 17 | 17 | 17 |
| MA | 50 nmol mol$^{-1}$ | NA | -46.13 ± 3.25 | -46.13 ± 3.25 | 0 | 16 | 16 |
| York | 50 nmol mol$^{-1}$ | -42.56 ± 3.54 | -39.99 ± 3.45 | -39.99 ± 3.34 | 14 | 1 | 1 |
| BCES Orthogonal | 50 nmol mol$^{-1}$ | -46.09 ± 0.56 | NA | NA | 27 | 0 | 0 |
| BCES (Y\|X) | 50 nmol mol$^{-1}$ | -42.98 ± 2.88 | -43.96 ± 2.73 | -43.95 ± 2.73 | 21 | 20 | 20 |
| OLS | 100 nmol mol$^{-1}$ | -41.90 ± 3.44 | -42.80 ± 3.39 | -42.80 ± 3.39 | 22 | 21 | 21 |
| OLS II | 100 nmol mol$^{-1}$ | -42.28 ± 3.33 | -42.98 ± 3.30 | -42.98 ± 3.30 | 17 | 17 | 17 |
| MA | 100 nmol mol$^{-1}$ | -32.27 ± 4.79 | -44.97 ± 3.23 | -44.97 ± 3.23 | 1 | 14 | 14 |
| York | 100 nmol mol$^{-1}$ | -42.24 ± 3.51 | -39.07 ± 4.54 | -39.07 ± 4.40 | 9 | 1 | 1 |
| BCES Orthogonal | 100 nmol mol$^{-1}$ | -46.00 ± 0.56 | NA | NA | 27 | 0 | 0 |



| | | | | | $n_{AirCore}$ Keeling method | $n_{AirCore}$ Miller-Tans 1 | $n_{AirCore}$ Miller-Tans 2 |
|---|---|---|---|---|---|---|---|
| BCES (Y\|X) | 100 nmol mol$^{-1}$ | -42.34 ± 2.76 | -43.39 ± 2.65 | -43.38 ± 2.65 | 22 | 21 | 21 |
| OLS | 10 s | -41.23 ± 3.22 | -42.39 ± 2.45 | -42.39 ± 2.45 | 21 | 19 | 19 |
| OLS II | 10 s | -40.18 ± 2.97 | -41.20 ± 2.09 | -41.20 ± 2.29 | 18 | 17 | 17 |
| MA | 10 s | -32.40 ± 2.36 | -43.75 ± 2.29 | -43.75 ± 2.29 | 2 | 17 | 17 |
| York | 10 s | -40.36 ± 3.93 | -40.13 ± 3.53 | -40.09 ± 3.70 | 9 | 4 | 5 |
| BCES Orthogonal | 10 s | -47.29 ± 0.49 | -34.04 ± 2.43 | -33.76 ± 3.56 | 28 | 4 | 8 |
| BCES (Y\|X) | 10 s | -41.45 ± 2.88 | -42.41 ± 2.36 | -42.40 ± 2.36 | 26 | 20 | 20 |
| OLS | 15 s | -41.19 ± 3.13 | -41.66 ± 2.73 | -41.66 ± 2.73 | 20 | 22 | 22 |
| OLS II | 15 s | -39.83 ± 3.09 | -40.46 ± 2.61 | -40.46 ± 2.61 | 18 | 20 | 20 |
| MA | 15 s | -7.58 ± 2.44 | -42.70 ± 2.37 | -42.70 ± 2.37 | 3 | 17 | 17 |
| York | 15 s | -40.41 ± 3.75 | -39.39 ± 3.48 | -39.53 ± 3.71 | 4 | 2 | 3 |
| BCES Orthogonal | 15 s | -47.28 ± 0.50 | -31.41 ± 3.00 | -34.50 ± 3.14 | 28 | 3 | 4 |
| BCES (Y\|X) | 15 s | -41.33 ± 3.12 | -42.21 ± 2.14 | -42.20 ± 2.14 | 27 | 21 | 21 |

**Table C.2 CRDS AirCore samples. $N_{AirCore}$ represents number of AirCore samples used to determine averaged $\delta^{13}CH_4$ after applying rejection criterium. $C_2H_6$ on $\delta^{13}CH_4$ correction applied.**

| Linear Fitting | Data Cluster | $\delta^{13}CH_4 \pm u(\delta^{13}CH_4)$ | | | $n_{AirCore}$ Keeling method | $n_{AirCore}$ Miller-Tans 1 | $n_{AirCore}$ Miller-Tans 2 |
|---|---|---|---|---|---|---|---|
| | | Keeling method | Miller-Tans method 1 | Miller-Tans method 2 | | | |
| OLS | raw | -43.30 ± 2.96 | -43.57 ± 1.61 | -43.57 ± 1.61 | 21 | 13 | 13 |
| OLS II | raw | -43.30 ± 2.99 | -43.57 ± 1.62 | -43.57 ± 1.62 | 21 | 13 | 13 |
| MA | raw | -22.88 ± 4.66 | -47.65 ± 1.83 | -47.65 ± 1.83 | 1 | 13 | 13 |
| York | raw | -43.63 ± 2.80 | -43.41 ± 2.89 | -43.28 ± 2.17 | 21 | 13 | 13 |
| BCES Orthogonal | raw | -47.94 ± 1.47 | -38.21 ± 2.37 | -42.36 ± 1.88 | 11 | 10 | 13 |
| BCES (Y\|X) | raw | -42.94 ± 2.58 | -43.59 ± 1.49 | -43.57 ± 1.49 | 23 | 13 | 13 |
| OLS | 10 nmol mol$^{-1}$ | -43.67 ± 2.83 | -44.56 ± 2.37 | -44.56 ± 2.37 | 19 | 18 | 18 |
| OLS II | 10 nmol mol$^{-1}$ | -43.67 ± 2.94 | -44.56 ± 2.47 | -44.56 ± 2.47 | 19 | 18 | 18 |





| | | | | | | | |
|---|---|---|---|---|---|---|---|
| MA | 10 nmol mol$^{-1}$ | NA | -47.95 ± 2.77 | -47.95 ± 2.77 | 0 | 18 | 18 |
| York | 10 nmol mol$^{-1}$ | -44.06 ± 3.19 | -44.05 ± 4.13 | -43.97 ± 4.17 | 18 | 3 | 4 |
| BCES Orthogonal | 10 nmol mol$^{-1}$ | -47.85 ± 0.55 | -3037 ± 1.79 | -30.94 ± 1.75 | 28 | 1 | 1 |
| BCES (Y\|X) | 10 nmol mol$^{-1}$ | -43.70 ± 2.67 | -44.58 ± 2.17 | -44.57 ± 2.17 | 21 | 18 | 18 |
| OLS | 50 nmol mol$^{-1}$ | -44.46 ± 3.02 | -45.24 ± 2.90 | 45.24 ± 2.90 | 17 | 17 | 17 |
| OLS II | 50 nmol mol$^{-1}$ | -44.46 ± 3.27 | -45.24 ± 3.14 | -45.24 ± 3.14 | 17 | 17 | 17 |
| MA | 50 nmol mol$^{-1}$ | NA | -47.57 ± 3.00 | -47.57 ± 3.00 | 0 | 14 | 16 |
| York | 50 nmol mol$^{-1}$ | -44.50 ± 3.54 | -42.75 ± 3.45 | -42.74 ± 3.34 | 14 | 1 | 1 |
| BCES Orthogonal | 50 nmol mol$^{-1}$ | -47.49 ± 0.58 | NA | NA | 27 | 0 | 0 |
| BCES (Y\|X) | 50 nmol mol$^{-1}$ | -44.81 ± 2.80 | -45.90 ± 2.69 | -45.89 ± 2.69 | 20 | 19 | 19 |
| OLS | 100 nmol mol$^{-1}$ | -43.82 ± 3.30 | -44.51 ± 3.33 | -44.51 ± 0.91 | 20 | 20 | 20 |
| OLS II | 100 nmol mol$^{-1}$ | -44.05 ± 3.44 | -44.85 ± 3.28 | -44.85 ± 3.28 | 17 | 16 | 16 |
| MA | 100 nmol mol$^{-1}$ | -41.53 ± 3.57 | -46.96 ± 3.32 | -46.95 ± 3.32 | 1 | 14 | 14 |
| York | 100 nmol mol$^{-1}$ | -44.58 ± 3.51 | -42.06 ± 4.54 | -42.05 ± 4.40 | 9 | 1 | 1 |
| BCES Orthogonal | 100 nmol mol$^{-1}$ | -47.39 ± 0.58 | NA | NA | 27 | 0 | 0 |
| BCES (Y\|X) | 100 nmol mol$^{-1}$ | -44.29 ± 2.64 | -45.10 ± 2.71 | -45.09 ± 2.71 | 21 | 21 | 21 |
| OLS | 10 s | -43.35 ± 3.21 | -44.29 ± 2.42 | -44.29 ± 2.42 | 21 | 19 | 19 |
| OLS II | 10 s | -43.40 ± 3.13 | -44.39 ± 2.34 | -44.39 ± 2.34 | 19 | 18 | 18 |
| MA | 10 s | -35.76 ± 3.56 | -45.77 ± 2.28 | -45.77 ± 2.28 | 2 | 17 | 17 |
| York | 10 s | -42.81 ± 3.95 | -42.48 ± 3.56 | -42.41 ± 3.73 | 9 | 4 | 6 |
| BCES Orthogonal | 10 s | -48.56 ± 0.48 | -36.77 ± 2.32 | -36.53 ± 3.55 | 28 | 4 | 9 |
| BCES (Y\|X) | 10 s | -43.54 ± 2.71 | -43.81 ± 2.21 | -43.79 ± 2.21 | 25 | 19 | 19 |
| OLS | 15 s | -43.23 ± 3.10 | -43.65 ± 2.70 | 43.65 ± 2.70 | 20 | 22 | 22 |
| OLS II | 15 s | -42.02 ± 3.09 | -42.70 ± 2.62 | -42.70 ± 2.62 | 18 | 20 | 20 |





| | | | | | | | |
|---|---|---|---|---|---|---|---|
| MA | 15 s | -20.48 ± 2.95 | -44.78 ± 2.34 | -44.78 ± 2.34 | 3 | 17 | 17 |
| York | 15 s | -42.21 ± 3.26 | -42.15 ± 3.50 | -42.16 ± 3.76 | 3 | 2 | 3 |
| BCES Orthogonal | 15 s | -48.53 ± 0.49 | -34.44 ± 2.88 | 36.78 ± 2.12 | 28 | 3 | 3 |
| BCES (Y\|X) | 15 s | -43.00 ± 2.92 | -44.82 ± 2.25 | -44.80 ± 2.25 | 25 | 22 | 22 |