# Peer review of "Statistical evaluation of methane isotopic signatures determined during near-source measurements"

_EGUsphere, 2023_

## Referee Comment (RC1)

The authors of the manuscript 'Statistical evaluation of methane isotopic signatures determined during near-source measurements' compare the effects from various measurement techniques and analytical methods on $\delta^{13}$C-CH$_4$ source values within a controlled gas release experiment. They intend to give generalized recommendations for the best practice in determing $\delta^{13}$C-CH$_4$ source values during mobile measurements.

In general, because of the controlled conditions, the experimental set-up provides a good basis for studying the effects of near-source measurements. However, I think that the set-up is more suitable for studies that focus on the comparison of measurement methods and the change of environmental conditions such as distance to the source or changes in fluxes.
Comparing different analysis methods (mass conservation method, linear fitting, averaging) is a good idea in principle and could be helpful in establishing best practice guidelines. However, the analysis methods observed here are potentially better suited for computational/modeling studies in which the dominant effects are studied in the context of a multivariate approach.
All in all, I would not recommend this manuscript for publication at the present stage due to the reasons addressed in the general and specific comments below:

general comments:

First of all, the structure of the manuscript is quite poor. For the reader, the content is often confusing and difficult to follow, as basic information, results, discussion and conclusion are wildly mixed in the different manuscript sections and also between the main part of the manuscript and the appendix (see specific comments). Especially because of the numerous analysis parameters to be considered, it is important to create a clear structure, so that the reader does not lose the overview. In this context, Figure 1 is very helpful. It should therefore be used as a "guide" through the manuscript. For example, the analytical methods as well as the results could be presented in the same order as shown in Figure 1 using the same titles, such as: 1. mass conservation method, 2. background subtraction, ... And, similar, for the results: 1. effects from mass conservation, 2. effects from different backgrounds, ...In addition, the various results sections should each contain a corresponding figure showing a comparison of one single parameter. This will contribute to greater clarity and conciseness. A minor point is the clear distinction between the observed $\delta^{13}$CH$_4$ and source $\delta^{13}$CH$_4$ values by using different indices such as 'o', 's'.

The title is somewhat misleading as it implies a dominance of statistical analysis. However, the manuscript describes more of an application study, investigating numerous parameters based on near-source laboratory measurements and comparing different analytical methods such as mass conservation method or linear fitting method. From the title, I would have expected a manuscript on multivariate statistics to understand the relationships between the numerous parameters and their importance in determining isotopic source values.

Another important criticism is that the novelty of the study is not obvious. The results presented show more or less similar results to previous studies. There are already several publications comparing the different linear fitting methods and mass conservation methods that conclude that the Miller-Tans-Method is the best method when the background value varies, and that the York method is the best fitting method for uncertainties in both, x as well as in y. Some of the results presented here are already self-evident, such as that CRDS uncertainties are higher than those of IRMS measurements due to lower instrument precision.

In my opinion, the requirements for a best-practice recommendation, which the authors have set as their goal, are not met. The given requirements for a generally valid analysis concept are not fulfilled, since the measurements are performed under very specific conditions (synthetic/laboratory) and

several problems have not yet been solved, such as the possible temperature effects of the direct measurements or the depletion of CRDS values compared to IRMS values (potential calibration effects).

specific comments:

Abstract:

- line 20: The issues 'accessibility, practicality and costs' are not addressed further in the manuscript. I would have expected these topics to be included in the main part as well.
- general: normally, the abstract is a brief summary of the research presented and therefore should also include the main findings of the study, which is what is missing here.

1.  Introduction:
    - line 58: To me, the phrase 'sampling over five days' implies a time series. Since you are working with an artificial source that should provide constant values over time, I would rather speak of 'five consecutive runs'.
    - line 62-67: this section rather belongs to the chapter conclusion/outlook

2.  Controlled release experiment and sampling methodology:
    - controlled release set up:
        o  what is the pressure and volume of the gas cylinders?
        o  at which location does the release experiment take place?
        o  is there a control of wind speed and fluxes?
        o  a more detailed description (besides mentioning of Gardiner et al.) of the controlled release system would be beneficial in this section, as it is the core of the experiment. A graphical scheme would also be helpful.
        o  Does the decreasing gas pressure of the 12 cylinders have an effect on the isotopic values, since we have sometimes observed a slight change in isotope values with decreasing gas pressure in our standard gas bottles?
    - line 68: I would give chapter 2 a more general title like 'experimental setup'. As the mass conservation method is also an analytical method I would include subchapter 2.4 into chapter 3
    - line 80: The use of '‰' to express isotope quantities is obsolete. It is no longer encouraged by IUPAC. According to Brand and Coplen (Assessment of international reference materials for isotope-ratio, 2014), the term 'milli-Urey' (mUr) should replace the old ‰ sign. See Werner and Cormier ('Isotopes-Terminology, Definitions and Properties', 2022) for current terminology.
    - line 83: why was the direct sample from the cylinders not taken both at the beginning and at the end of the experiment to rule out effects such as fluctuating gas pressure?
    - line 85: IRMS is not yet explained- please provide a short sentence on instrument specification or a link to the corresponding chapter.
    - line 91-105: here the link to Appendix A is missing, where the methods are explained in detail.
    - line 110: did I understand this correctly: the Picarro data from the RHUL mobile laboratory was not used for continuous measurement analysis, but only for peak detection for the bag samples? Why not using here an AirCore sampling technique as well?

3.  Analytical methods of the acquired measurements
    - line 160-162: I do not understand that sentence.

- line 162-165: why do you not specify a wide range of different random backgrounds for sensitivity analysis instead of using 'average', 'global' and 'random' backgrounds, some of which are very close to each other?
- line 181: define 'R' for the reader who is not familiar with programing languages.
- line 181: what is the mathematical difference between OLS II and MA, since you use the same command in R (lmodel2())?
- line 199: Does it even make sense to test a method other than the one that takes into account the error in x and y such as York and BCES-regression method? Because that is what you would expect from your data which is biased in x as well as in y.
- subchapter 3.5: For a better overview, this subchapter should definitely be placed at the beginning of Chapter 3.

4. Results
- line 268-270: A reference for that observation is missing? Table 1?
- line 272: I do not think, that the results are the same. I would rather speak of 'similar results'.
- line 288-295: this section belongs to the chapter discussion/conclusion
- line 299-300: I did not find information on temperature instability and cavity pressure in Appendix B
- line 301-302: would it be not more appropriate to compare the AirCore values with the CRDS values from Table 2, as both belong to the same method? Or, if you want to use a reference value, why not use those values measured directly from the cylinder because this is more or less the 'true source' of $\delta^{13}$C-values?
- line 311 ff: this paragraph is an interpretation and should not be part of the results section
- line 316-321: here, a figure would be helpful to emphasize the results.
- line 321,330 ff: recommendations should be no part of the results section.
- line 339 ff.: this is again an interpretation
- line 351-354: this information is redundant, as it can already be found in subchapter 2.2.
- line 368-369: I do not believe that temperature effects play such a large role in the observed isotope discrepancies. Since direct and indirect measurements are based on completely different conditions (differences in fluxes, flows, mixing effects), I would not have expected unbiased values (deviation from the 'true' direct measurements) for the indirectly measured isotope values.

5. Discussion
- line 384: The analytical technique used depends primarily on the environmental conditions and requirements, and on the sensitivity required (relative or absolute measurements). For field measurements, it is not always possible to take bag samples and store them for IRMS measurements.
- line 388: The Miller-Tans-method should be used in any case where the background varies (usually in long-term studies). If the background is constant or not measured, the Keeling method can be used. Accordingly, there is not really a question which of the two methods is the better one, because the application depends on measurement conditions.
- line 390: Hoheisel et al. (2019) seem to come to different conclusions: "*Especially for natural gas samples, the precise determination and correction of C2H6 is important as in our study C2H6 can bias 13CH4 by up to 3‰ depending on the CH4-to-C2H6 ratio of the sample and the calibration cylinder.*" How can you explain this discrepancy?
- line 400-402: there are some more studies for example Takriti et al. ('*Mobile methane measurements: Effects of instrument specifications on data interpretation, reproducibility, and*

*isotopic precision',* 2021) and Hoheisel et al. (2019). So why are you focusing only on the studies mentioned in line 401-402 to compare your results?

- line 415: not only Wehr and Saleska proposed York fitting method, but also many other studies come to the same conclusion such as for example Hoheisel et al. (2019)
- line 420-422: Wehr and Saleska performed model simulations within their study. It is difficult to compare this result with the results of a field test conducted as part of this study.
- line 424-427: what is the conclusion from that statement? Instrument settings have not been discussed before, and introducing another parameter at this point is inappropriate. Therefore, I see no benefit in mentioning this here.
- line 428-429: this is a self-evident fact that applies to all occasions that require sample dilution.
- line 430: I would rather speak of expectable then remarkable (see last comments on results). In general, possible fractionation effects should be checked before starting the actual experiment: e.g. due to different wind speeds, different inflow angles, different temperatures etc. As long as the experimental framework conditions are not clarified, an experiment that investigates numerous parameters is not useful, in my opinion.
- line 436-439: I consider this statement questionable as the experimental conditions are very specific (artificial release of gases to the atmosphere).

6. Conclusion
   - line 447: I would not speak of statistical methods, but rather of a comparison under different methodological and analytical aspects

Appendix

- Appendix B should appear in a shortened version in the main part of the manuscript (one to two sentences)
- At the end of Appendix B a title ('Results for bag samples measured on IRMS and CRDS for all examined linear fitting methods') appears without text following
- Appendix C: A figure would help to clarify the impact of data clustering. I would include this section in the main part of the manuscript.
- for the mobile sampling laboratories, a table with specifications would be beneficial for a better overview

Figures and Tables

- Figure 1: The figure is very clear and helpful to the reader. However, I miss the applied mobile laboratory and the measured components ($CH_4$, $\delta^{13}CH_4$) as a note in the graph
- Figure 1: inconsistency: the averaged AirCore samples are presented in ppb- in the main part they are given in nmol/mol
- Figure 3: plotting different $CH_4$ mole fractions based on the size of the data points is absolutely confusing - rather use an average value. A general advice: do not show more than one dependency in one figure
- Figure 3: what does 'release' mean? Does this refer to the day of release or to the run number?
- Figure 3: Do you really want to include the data from release 4-5, which is, as you mentioned in the text, biased by calibration errors? Since you also omit the data for averaging I would not show it at all as it confuses the reader.
- Figure B1, C1: these figures are not mentioned anywhere in the text!
- Figure C1: use similar y-axis for better comparison
- Table 3: probably transposed numbers for MA- $\delta^{13}CH_4$ Keeling method? (-42.18 instead of -24.18?)

technical corrections:

- please check again for correct super/subscript (i.e., CH4 instead of $CH_4$) throughout the whole manuscript as there are many discrepancies
- line 130 ff: use consistently the term $CH_{4s}$ for source values to distinguish source values from observed values (see formula 1)
- line 31: delete the word 'measurements'
- line 46: better write 'equipped with an atmospheric sampling system (AirCore)' as you have not introduced the AirCore system yet
- line 51: 'determine $\delta^{13}C$ **source** values'
- line 146: delete commas in that sentence
- line 265-267: ensure that the formulas are displayed correctly: $\mu mol \cdot mol^{-1}$
- line 268: space between treatmen t
- line 339-340: rewrite the sentence as it is not clear (3 times the word corrections!)